# A New Kind of Network? Review and Reference Implementation of *Neural Cellular Automata*.

**Martin Spitznagel**                                          *martin.spitznagel@hs-offenburg.de*
*Institute for Machine Learning and Analytics (IMLA)*
*Offenburg University, Germany*

**Janis Keuper**                                              *janis.keuper@hs-offenburg.de*
*Institute for Machine Learning and Analytics (IMLA)*
*Offenburg University, Germany*
*& University of Mannheim, Germany*

**Reviewed on OpenReview:** *https://openreview.net/forum?id=NRwjjOZLqO*

## Abstract

*Stephen Wolfram* proclaimed in his 2003 seminal work *"A New Kind Of Science"* that simple recursive programs in the form of *Cellular Automata* (CA) are a promising approach to replace currently used mathematical formalizations, e.g. differential equations, to improve the modeling of complex systems. Over two decades later, while *Cellular Automata* have still been waiting for a substantial breakthrough in scientific applications, recent research showed new and promising approaches which combine *Wolfram's* ideas with learnable Artificial Neural Networks: So-called *Neural Cellular Automata* (NCA) are able to learn the complex update rules of CA from data samples, allowing them to model complex, self-organizing generative systems.

The aim of this paper is to review the existing work on NCA and provide a unified modular framework and notation, as well as a reference implementation in the open-source library **NCAtorch**.

Project Website: `https://www.neural-cellular-automata.org/`
Source Code: `https://github.com/mspitzna/NCAtorch`

## 1 Introduction

### 1.1 Cellular Automata

*Cellular Automata* (CA) have been a subject of study since the 1940s, notably by *John von Neumann* (Von Neumann et al., 1966) and others. At its core, *Cellular Automata* are discrete computational models, which consist of a finite, regular grid $\mathcal{G}$ of discrete cells. Each cell holds a discrete state $s_{i,t}$ at a given discrete time-step $t$. As time progresses, cell states are updated recursively and synchronously across all cells, based on a set of rules $\mathcal{R}$ that utilize the local neighborhood $\mathcal{N}(s_{i,t})$ to calculate the new cell state: $s_{i,t+1} := \mathcal{R}[\mathcal{N}(s_{i,t})]$ (Schiff, 2011). Figure 1 shows the computation process and

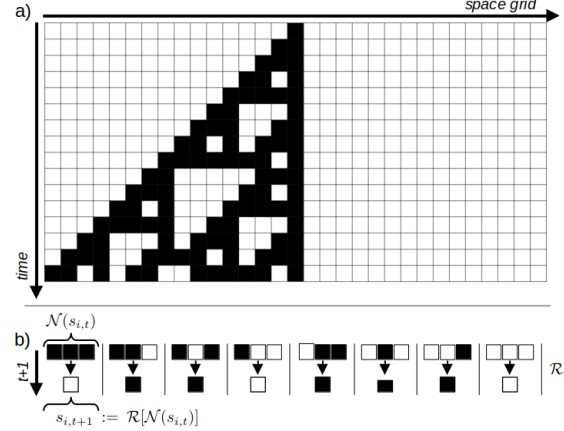

Figure 1: Visualization of a simple, 1D CA with binary states. a) shows the recursive update of grid states over time by applying the update rules $\mathcal{R}$ on the $3 \times 1$ cell neighborhoods shown in b).

rules for a basic CA with binary states ($s_{(x,y),t} \in \{0,1\}$).

*Cellular Automata* gained significant public recognition in the 1970s through *Conway's "Game of Life"* (Conway et al., 1970). This game employed a binary *Cellular Automaton* on a two-dimensional grid and update-rules based on a $3 \times 3$ neighborhood.

**Applications and Theoretical Significance.** Subsequent research demonstrated the broad applicability of CAs, including their use in *biological* (Coombes, 2009; Bouligand, 1986; Hatzikirou et al., 2012), *chemical* (Gerhardt & Schuster, 1989), and *physical modeling* (Wolfram, 1983; Załuska-Kotur et al., 2021). Furthermore, certain CA configurations were shown to possess powerful theoretical computing properties, such as *Turing Completeness* (Cook et al., 2004) of certain CA configurations, thus establishing CAs as a *universal computing model.*

This theoretical strength led *Stephen Wolfram* to propose his well-known work, ***"A New Kind Of Science"*** (Wolfram & Gad-el Hak, 2003). In it, he suggested that inherently limited mathematical formulations could be replaced by more powerful *"simple programs"* and formulated a new formal framework for numerous scientific applications grounded in *Cellular Automata.*

## 1.2 Basic *Neural Cellular Automata*

**From Fixed Rules to Neural Networks.** *Traditional Cellular Automata (CA)*, as initially proposed by (Wolfram & Gad-el Hak, 2003), (Conway et al., 1970), and (Von Neumann et al., 1966), are constructed using a fixed set of **manually designed rules**. For instance, the update rule shown in fig. 1 is known as *"Rule #110"* (named after the binary coding of the rule outputs), which is one of $2^{2^3} = 256$ possible rules for a 1D, binary-state CA with a neighborhood size of 3, first introduced in (Wolfram & Gad-el Hak, 2003).

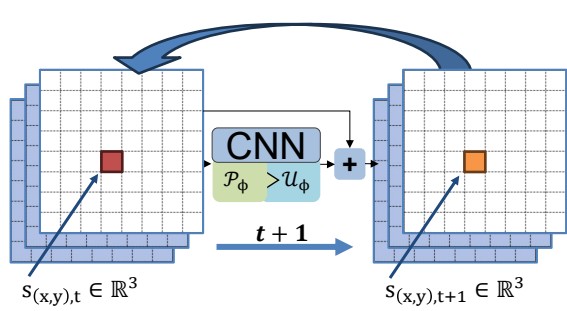

$s_{(x,y),t} \in \mathbb{R}^3$          $s_{(x,y),t+1} \in \mathbb{R}^3$

Figure 2: Sketch of a basic CNN implementation of a 2D NCA with a 3D state space. The initial grid is fed into a CNN architecture which updates the state additively and is called recursively for each time step. During training, the network is trained via usual gradient updates computed per timestep.

While all potential rules for a specific CA with predetermined discrete states $\mathcal{S}$ and neighborhood-size $|\mathcal{N}|$ can be generated combinatorially, these rule-spaces unfortunately **grow exponentially**. Even the relatively simple 2D binary CA used in the *Game of Life* already has $2^{2^{3\times3}} = 2^{512}$ possible rules for its $3 \times 3$ neighborhood. This challenge has made the idea of **learning more complex rules from data** — instead of designing them by hand — increasingly appealing.

Mordvintsev et al. (2020) introduced the concept of *Neural Cellular Automata* (NCA), which essentially substitutes the manually designed rules with artificial Neural Networks that are trained on problem-specific data. Equation (1) provides an abstract formalization of an NCA update function, where $f_\phi$ denotes an arbitrary neural network with learnable parameters $\phi$.

$$s_{i,t+1} := s_{i,t} + f_\phi[\mathcal{N}(s_{i,t})], \quad s_i \in \mathcal{S} \qquad (1)$$

Since $f_\phi$ is **space-invariant** with respect to the grid $\mathcal{G}$ (meaning the same $f_\phi$ is applied at all positions $i$ during a time step), (Mordvintsev et al., 2020) suggested an efficient implementation using "nested" *Convolutional Neural Networks (CNNs)* (LeCun & Bengio, 1998). In its simplest form, the kernels (or filters) of a single convolutional layer model a **learnable perception** $\mathcal{P}_{\mathcal{N},\phi}$ of neighborhoods $\mathcal{N}$ matching the kernel size, followed by a **learnable update layer** $\mathcal{U}_\phi$ implemented as $1\times1$ convolutions, which together implement learnable non-linear CA update rules $\mathcal{R}_\phi$ on all cells $s$ at time $t$ simultaneously:

$$s_{t+1} := s_t + \mathcal{R}_\phi[s_t] := s_t + \mathcal{U}_\phi[\mathcal{P}_{\mathcal{N},\phi} * s_t] \qquad (2)$$

Figure 2 illustrates this basic CNN-based NCA architecture. The theoretical equivalence between "nested" CNNs and CA has been formally proven by (Gilpin, 2019). It is important to note, however, that most

NCA architectures relax the original CA property of discrete cell states, moving instead toward **continuous vector state representations** $s_{i,t} \in \mathbb{R}^n$.

While the original work of Mordvintsev et al. (2020) used a simple, *LeNet* (LeCun et al., 2002)-like, 1-layer convolutional NCA to show impressive self-organizing and classification tasks on images, later works extended the use of CNNs to more complex network architectures: Sandler et al. (2020) showed NCA based image segmentation with larger CNNs, while Tesfaldet et al. (2022a) introduced an *attention* based implementation. NCAs based on GANs (Otte et al., 2021a), VAEs (Palm et al., 2022) or diffusion (Kalkhof et al., 2024a) also have been utilized for generative tasks. Grattarola et al. (2021) applied a NCA based approach on graph data.

### 1.3 Contributions

The contributions of this paper are threefold: I) We summarize the existing works on NCA in the (to the best of our knowledge) first systematic review and experimental evaluation of this novel research topic. II) We introduce a unified notation and modular decomposition that consolidates the disparate formulations of prior NCA works into a single coherent framework. III) We release ***NCAtorch***[1], a *PyTorch* (Paszke et al., 2019) open-source framework that consolidates established NCA practices and, novel to this domain, makes a range of standard deep-learning stability and training options available as configuration-level components in a single ready-to-use stack.

## 2 Reference Implementation: the *NCAtorch* Framework

To facilitate systematic and reproducible research in NCA, we introduce a modular and extensible framework for experimentation. Its central objective is to compare different NCA architectures and components across a spectrum of tasks. The framework is built around an abstract base model, with key components managed through YAML files and *pydantic* for type-safe configuration. This design allows researchers to seamlessly implement and integrate new perception modules, training routines, and datasets. To ensure traceability, all training metrics and visualizations are optionally logged via Weights & Biases (Biewald, 2020).

**Related Frameworks.** The original NCA and self-organizing systems releases are distributed as independent example notebooks without a shared configuration or evaluation scaffold (Mordvintsev et al., 2020; Randazzo et al., 2020; 2021; Niklasson et al., 2021). More recent libraries provide reusable NCA implementations: *ncalib*[2] offers a modular Python library with example NCA variants (e.g., growing, self-classifying MNIST, segmentation, genome-based NCAs), while CAX[3] (Faldor & Cully, 2025) provides a JAX-based library for accelerated cellular systems that includes NCA variants alongside a broader collection of artificial-life models.

**What NCAtorch adds.** NCAtorch is a PyTorch-centered research framework for controlled, cross-task comparisons of NCA variants. It provides a unified training and evaluation stack with standardized configuration, shared registries for models/datasets/trainers, and logging/visualization utilities, enabling reproducible ablations across vision tasks. The framework is ready to use out of the box and includes documentation for extending it with new datasets, perception modules, and update rules. Beyond reusable NCA implementations, NCAtorch makes a range of commonly used training and stability options readily accessible in the NCA setting. Specifically, it supports NCA-established mechanisms such as stochastic updates (fire rate), living masks, and sample pooling with perturbations (Mordvintsev et al., 2020; Randazzo et al., 2020; 2021; Niklasson et al., 2021), as well as standard deep-learning engineering features including learning-rate scheduling, mixed precision, gradient clipping/accumulation, and checkpointing. Importantly, these components are exposed through a unified, modular interface with swappable perception and update modules, and can be enabled or disabled via YAML configuration to support controlled comparisons.

**Modular Design Philosophy.** Following the general NCA formulation $\mathbf{s}_{t+1} = \mathbf{s}_t + \mathcal{U}_\phi[\mathcal{P}_\phi(\mathbf{s}_t)]$ from eq. (2), our framework architecture is organized into four core components that can be independently configured and combined: (1) *Cell State Representation* $\mathbf{s}_t \in \mathbb{R}^{H \times W \times C}$: defining what information each cell stores and

---

[1] https://github.com/mspitzna/NCAtorch
[2] https://github.com/dwoiwode/ncalib
[3] https://github.com/maxencefaldor/cax

propagates across the $H \times W$ grid with $C$ channels. (2) learnable *Perception Module* $\mathcal{P}_\phi$: determining how cells sense their local neighborhood and extract spatial features. (3) learnable *Update Module* $\mathcal{U}_\phi$: computing state transitions from perceived features, and (4) *Training Infrastructure*: managing optimization, logging, and specialized training strategies such as Sample Pooling. Figure 3 visualizes the interaction of these components within a single NCA iteration step. This separation of concerns serves two critical purposes. First, it enables *systematic experimentation*: by keeping the update module, training protocol, and cell state structure constant while varying only the perception module, we can isolate and quantify the impact of different perception strategies on learning dynamics, generation quality, and robustness (section 3). Second, it promotes *extensibility*: new perception types, loss functions, or training mechanisms can be integrated as drop-in replacements without modifying the core architecture. The framework thus functions both as a research tool for controlled comparative studies and as a flexible platform for exploring novel NCA designs.

## 2.1 Modular CA Model and Perceptions

**Cell State Representation.** At the core of our approach is a highly adaptable CA model with a customizable cell state representation. Each cell's state vector can be structured to contain: (i) *visible channels* for observable outputs (e.g., RGB values for image generation), (ii) *hidden channels* for internal computation and information propagation, (iii) *condition channels* for fixed global or local conditioning signals (e.g., target class labels, spatial guidance), and optional (iv) *classification channels* for evolving class predictions in recognition tasks. This flexibility allows the same architectural foundation to be adapted across generation, classification, and video prediction tasks (section 3). During each forward pass, the evolving state channels are concatenated with any auxiliary condition channels (e.g., class labels broadcast spatially or edge maps for conditional generation) to prepare the input for the perception module, as illustrated in fig. 3.

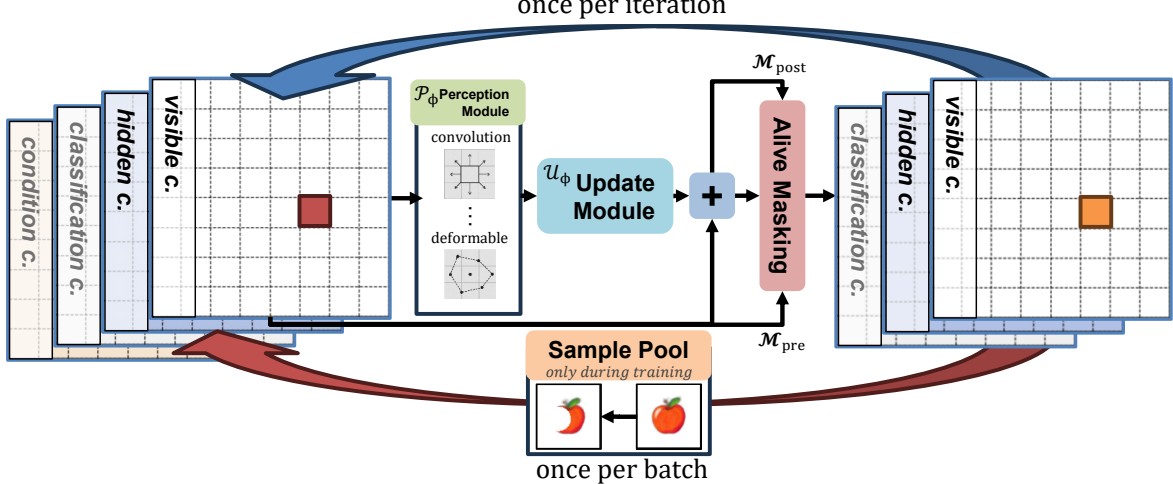

Figure 3: Overview of a single NCA iteration step. The cell state consists of visible channels (e.g., RGB), hidden channels for internal representations, and optional classification or conditioning channels. The perception module processes the cell state using one of several configurable options. The update module processes the perceived features to compute state updates. Stochastic updates and alive masking are applied before updating the visible, hidden, and optional classification channels to form the new cell state for the next iteration. The sample pool is used after one batch of images is processed to retain and reuse evolved states for subsequent training iterations.

**Perception Module.** The perception module (fig. 3) functions as the sensory component of the CA, extracting local spatial features from the cell state grid. Critically, the framework supports a wide array of interchangeable perception strategies as independent modules, enabling systematic comparative evaluation of their impact on learning dynamics and robustness. We provide implementations of five distinct perception mechanisms: (i) *Convolution Perception*: standard convolution with configurable perception size (kernel size) and dilation rate. (ii) *Attention Perception* (Tesfaldet et al., 2022b): self-attention with adjustable perception

radius. (iii) *Sobel Perception*: gradient-based edge detection with fixed $3 \times 3$ kernel. (iv) *Deformable Perception*: deformable convolution with learnable spatial offsets and configurable perception size (Dai et al., 2017) and (v) *Residual Perception*: residual convolution with two sequential convolutional layers and skip connections, with configurable perception size. All configurable perception modules expose perception size as a hyperparameter, facilitating controlled experimentation on the influence of receptive field scope on emergent behavior.

**Update Module.** The update module (fig. 3) computes the incremental state change $\Delta\mathbf{s}$ based on features extracted by the perception module. It is implemented as a configurable sequence of convolutional layers with non-linear activations (e.g., ReLU), defaulting to a single depthwise convolution for computational efficiency. After computing the update, a stochastic update mask is applied for asynchronous updates and, optionally, a living mask. The masked update is then aggregated with the current state to complete one iteration.

**Living Mask Mechanism.** To enable robust pattern regeneration and boundary formation, our framework implements an optional living mask (fig. 3) that enforces cell viability constraints during state evolution (Mordvintsev et al., 2020). The mechanism operates on a configurable alpha channel within the cell state. A cell is considered alive if its alpha value exceeds a threshold $\tau = 0.1$. The living mask is computed both before and after the state update, and only cells that remain alive in both states are allowed to persist: $\mathcal{M} = \mathcal{M}_{\text{pre}} \wedge \mathcal{M}_{\text{post}}$. This combined mask is applied element-wise to zero out dead cells: $\mathbf{s}_{t+1} = \mathbf{s}_{t+1} \times \mathcal{M}$.

## 2.2 Training Algorithm

Unlike standard supervised models that compute a loss after a single forward pass, NCA training *unrolls* the update rule for $T$ sequential steps before evaluating the loss, and backpropagates gradients through all steps. While the core rollout mechanism is shared across all NCA tasks, the concrete training setup varies by objective (see section 3 for task-specific details and algorithms 2 and 3 for variant pseudocode). Here we present the general procedure for image generation. Each training sample consists of a seed state $\mathbf{x}_{\text{seed}}$ (e.g., an input image), an optional condition $\mathbf{c}$ (e.g., one-hot class label), and a target image $\mathbf{x}_{\text{target}} \in \mathbb{R}^{H \times W \times C_v}$, where $C_v$ denotes the number of visible output channels (e.g., 4 for RGBA). The initial state $\mathbf{s}_0 \in \mathbb{R}^{H \times W \times C}$ with $C = C_v + C_h + d_c$ is formed by placing the seed into the $C_v$ visible channels, initializing the $C_h$ hidden channels to zero, and broadcasting the $d_c$-dimensional condition across all spatial positions. After $T$ steps, the visible channels $\hat{\mathbf{x}} = \mathbf{s}_T[: C_v]$ are compared to $\mathbf{x}_{\text{target}}$ via a loss $\mathcal{L}$ (e.g., MSE). The rollout length $T \sim \mathcal{U}[T_{\min}, T_{\max}]$ is randomized per iteration for temporal regularization. A sample pool (section 2.4) may replace the initial state with a pre-

---

**Algorithm 1:** NCA Training for Generative Image Tasks

**Input** : Training sample $(\mathbf{x}_{\text{seed}}, \mathbf{c}, \mathbf{x}_{\text{target}})$, NCA model $\mathcal{F}_\phi$ (perception $\mathcal{P}_\phi$, update $\mathcal{U}_\phi$), loss $\mathcal{L}$, step range $[T_{\min}, T_{\max}]$, $p_{\text{fire}}$

**Output:** Trained parameters $\phi^*$

1 Initialize optimizer, LR scheduler, sample pool $\mathcal{P}_{\text{pool}} \leftarrow \emptyset$;

2 **for** *each training iteration* **do**

3     Draw $(\mathbf{x}_{\text{seed}}, \mathbf{c}, \mathbf{x}_{\text{target}})$ from dataset $\mathcal{D}$;

4     Construct initial state: $\mathbf{s}_0 \leftarrow \texttt{embed}(\mathbf{x}_{\text{seed}}, \mathbf{c})$

    `// Sample Pool`

5     **if** $|\mathcal{P}_{\text{pool}}| > 0$ **then**

6         With probability $r_{pool}$: replace $\mathbf{s}_0$ with a random sample from $\mathcal{P}_{\text{pool}}$;

7         Optionally apply spatial damage or condition mutation;

    `// Multi-step NCA rollout`

8     Sample rollout length $T \sim \mathcal{U}[T_{\min}, T_{\max}]$;

9     **for** $t = 0$ **to** $T - 1$ **do**

        `// perception: local features`

10         $\mathbf{p}_t \leftarrow \mathcal{P}_\phi(\mathbf{s}_t)$;

        `// update: state increment`

11         $\Delta\mathbf{s}_t \leftarrow \mathcal{U}_\phi(\mathbf{p}_t)$;

        `// stochastic update mask`

12         $\mathbf{m}_t \sim \text{Bern}(p_{\text{fire}})^{H \times W}$;

13         $\mathbf{s}_{t+1} \leftarrow \mathbf{s}_t + \mathbf{m}_t \odot \Delta\mathbf{s}_t$;

        `// living mask (optional)`

14         $\mathbf{s}_{t+1} \leftarrow \mathbf{s}_{t+1} \cdot \mathcal{M}(\mathbf{s}_t, \mathbf{s}_{t+1})$;

    `// extract visible channels as prediction`

15     $\hat{\mathbf{x}} \leftarrow \mathbf{s}_T[: C_v]$;

16     $L \leftarrow \mathcal{L}(\hat{\mathbf{x}}, \mathbf{x}_{\text{target}})$;

17     Backpropagate $\nabla_\phi L$ through all $T$ steps; clip gradients; optimizer step;

    `// Commit evolved state to pool`

18     Store $(\mathbf{s}_T, \mathbf{c}, \mathbf{x}_{\text{target}})$ in $\mathcal{P}_{\text{pool}}$ (FIFO);

viously evolved state, enabling the model to learn self-repair and long-term stability. Algorithm 1 provides the complete pseudocode.

### 2.3 Trainer Framework and Training Strategy

Our training framework features an abstract base trainer that encapsulates core functionalities such as optimizer initialization (e.g., AdamW), learning rate scheduling (e.g., cosine annealing), checkpointing, and comprehensive logging, including monitoring of intermediate evolution steps (fig. 4). The base trainer handles technical aspects like mixed precision, gradient clipping, gradient accumulation, and optional gradient checkpointing for memory efficiency.

Specialized trainers can be derived from this base to target specific objectives. Functionalities such as latent space evolution are integrated at the model level for broader applicability.

### 2.4 Sample Pool Mechanism

To enhance training dynamics and stability, particularly for generative tasks, our framework incorporates the sample pool mechanism (Mordvintsev et al., 2020). This module retains and reuses representative samples from previous iterations, as visualized in fig. 4, which are reintroduced into training batches based on a configurable ratio. To increase robustness, the sample pool employs a controlled perturbation process, such as applying a masking function to reintroduce damaged samples. This mechanism helps balance exploration and exploitation, contributing to a more robust training process for NCAs. A detailed empirical ablation study demonstrating the impact of Sample Pooling on regeneration performance is provided in appendix A.4.

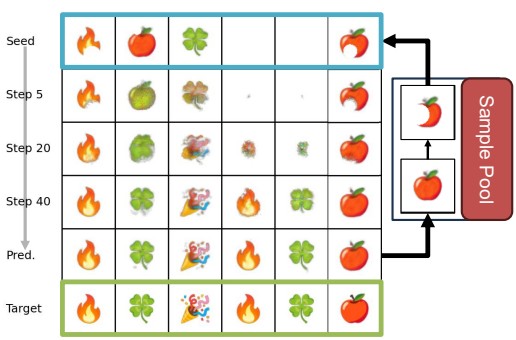

Figure 4: Sample pooling mechanism during training. Starting from seed states (top row), the NCA evolves over multiple steps toward target patterns (bottom row). Optional intermediate logging states show the progressive refinement. The sample pool retains evolved states from previous training iterations, which are randomly selected and reintroduced into new training batches, optionally with applied perturbations or mutations.

### 2.5 Visualization Toolkit

Our visualization toolkit provides an interactive interface to explore and evaluate trained CA models. Built on a FastAPI backend with Jinja2 templates, it enables real-time observation of the model's state evolution in a web browser. The toolkit supports dataset-specific functionalities, such as painting MNIST digits or dynamically changing targets for emoji generation to test regeneration. This interactivity facilitates detailed qualitative analysis and an intuitive understanding of the CA dynamics. An example of the visualization interface is shown in fig. 5.

## 3 Evaluation of Neural Cellular Automata

We present a systematic evaluation of existing NCA work, extending prior approaches with additional

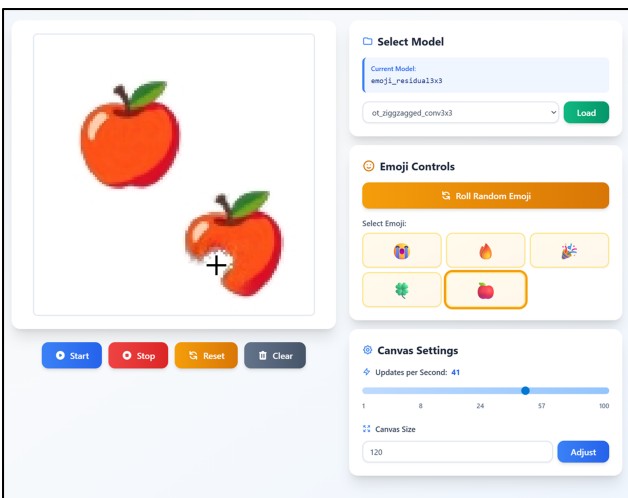

Figure 5: Interactive web-based visualization toolkit for real-time NCA model inference and exploration.

enhancements to perception mechanisms and train-
ing protocols. Our experiments are designed to rigorously evaluate the proposed framework, with a particular
focus on the impact of the perception module on NCA performance. By systematically varying the percep-
tion mechanism, we isolate its influence on learning dynamics, emergent behavior, and robustness. A key
aspect of our evaluation is quantifying the NCA's ability to regenerate target structures after significant
perturbations, such as targeted cell removal (**damage**) or random condition alterations (**mutation**). This
assessment allows us to measure the resilience inherent in different perception designs.

Our evaluation spans **image generation** (section 3.1), including conditional emoji generation, Edge-to-
Handbag translation, latent space NCAs and texture generation (section 3.2) with style-based synthesis on
the DTD dataset. **classification** (section 3.3) via self-classifying NCAs on MNIST and CIFAR-10 and **video
transition** (section 3.4) with temporal dynamics learning on MovingMNIST.

### 3.1 Image Generation (Emoji, E2H, MNIST)

In the image generation tasks, the objective is to learn NCA update rules that grow a target image from
a minimal seed state, following the training procedure in algorithm 1. The NCA evolves an initial state $\mathbf{s}_0$
for $T$ steps, and the loss $\mathcal{L}$ is computed between the predicted visible channels $\hat{\mathbf{x}} = \mathbf{s}_T[: C_v]$ and the target
image. We assess the ability of NCAs equipped with different perception modules to grow and maintain
specific target images. The modular perception architecture (section 2) enables a systematic comparison
of Sobel filters, standard convolutions ($3\times3$, $5\times5$), deformable convolutions, and MultiPerception strategies
while keeping all other components constant. Our experiments focused on two primary tasks: generating
static Emoji images (Mordvintsev et al., 2020) and performing image-to-image translation with the Edge-
to-Handbag (E2H) dataset (Isola et al., 2018). Additional experiments on growing MNIST digits and the
impact of different loss functions are presented in appendix A.5.

#### 3.1.1 Regeneration Metric

To quantify regeneration performance, we use a unified metric that measures recovery following a perturba-
tion. We define the recovery percentage as:

$$\text{Recovery (\%)} = \frac{m_{\text{final}} - m_{\text{d}}}{m_{\text{pre}} - m_{\text{d}}} \times 100 \tag{3}$$

where $m_{\text{pre}}$ is the mean task-specific metric before the perturbation, $m_{\text{d}}$ is the metric immediately after
the perturbation is applied, and $m_{\text{final}}$ is the metric at a defined final timestep post-perturbation. This
normalization yields 100% for complete recovery to the pre-perturbation performance level and 0% for no
improvement. Negative values signify further degradation after the initial perturbation.

For generation tasks, we report *Learned Perceptual Image Patch Similarity* (LPIPS) (Zhang et al., 2018), a
perceptual distance metric that computes a weighted $\ell_2$ distance between deep feature activations $\phi_l$ of a
pretrained network:

$$\text{LPIPS}(\hat{\mathbf{x}}, \mathbf{x}) = \sum_l \frac{1}{H_l W_l} \sum_{i,j} \|\mathbf{w}_l \odot (\phi_l(\hat{\mathbf{x}})_{i,j} - \phi_l(\mathbf{x})_{i,j})\|_2^2, \tag{4}$$

where $\mathbf{w}_l$ are learned channel-wise weights. Lower LPIPS indicates higher perceptual similarity to the target.

**Perturbation Types.** We evaluate robustness under two types of perturbations applied to an evolved state
$\mathbf{s}_t$ at a fixed timestep:

- **Damage**: a circular mask $\mathbf{M} \in \{0, 1\}^{H \times W}$ is applied to zero out a contiguous circular region of
  cells: $\mathbf{s}_t \leftarrow \mathbf{s}_t \cdot \mathbf{M}$. This tests the NCA's ability to regrow missing structure from surviving cells.

- **Mutation**: the condition channel $\mathbf{c}$ is replaced with a randomly sampled target class $\mathbf{c}' \neq \mathbf{c}$, while
  the cell state is kept intact. This tests the NCA's ability to adapt an already-evolved state to a new
  target.

### 3.1.2 Emoji Task

**Prior Work.** The original Growing CA work by Mordvintsev et al. (2020) demonstrated that NCAs could learn to grow and maintain a single target emoji image from a seed pixel using simple convolutional perception and MSE loss. A key innovation was the introduction of the *Sample Pooling* mechanism, where training batches included samples drawn from previously generated states rather than always starting from a clean seed. This technique significantly improved the model's ability to recover from perturbations and maintain stable patterns over extended evolution periods (Randazzo et al., 2021).

**Our Extension.** We extend this foundational work in three key directions. First, we implement *conditional multi-class generation* by adding a dedicated condition dimension to the cell state, as visualized in fig. 6. With this, a single NCA model is able to generate and transition between multiple emoji classes (we use 5 distinct emojis). This is achieved by encoding the target emoji as a one-hot vector that is spatially broadcast across all cells, providing a global conditioning signal. Second, we systematically evaluate the impact of different perception modules on both generation quality and robustness, comparing Sobel filters, standard convolutions (3x3, 5x5), deformable convolutions, and a Combined MultiPerception strategy. Third, we enhance the training protocol to explicitly promote adaptation capabilities: during training, we not only apply spatial damage to pooled samples but also randomly *mutate* the target emoji condition, forcing the NCA to learn transitions between different target patterns.

**Training Objective.** Following algorithm 1, the emoji task uses MSE loss $\mathcal{L}(\hat{\mathbf{x}}, \mathbf{x}_{\text{target}}) = \|\hat{\mathbf{x}} - \mathbf{x}_{\text{target}}\|^2$ between the predicted RGBA channels ($C_v = 4$) and the target emoji. Sample pooling with both damage and condition mutation is enabled to promote regeneration and multi-class adaptation.

**Results.** We trained NCAs with the setup described above, comparing five distinct perception strategies: Sobel filters, standard 3x3 and 5x5 convolutions, 3x3 attention, deformable 3x3 convolutions, and a 'Combined' strategy using MultiPerception with 3x3 convolution and 5x5 dilated convolution (dilation=2). Table 1 reports the quantitative results, including LPIPS loss at different stages of evolution (averaged over timesteps 64–96, at timestep 149, and at timestep 449) as well as the final regeneration percentages after controlled damage and mutation.

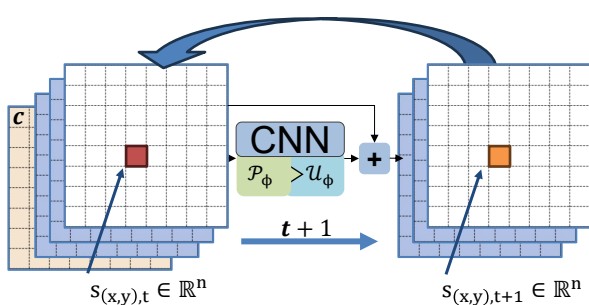

Figure 6: Architecture of conditional NCA. The cell state includes visible RGB channels, hidden channels, and a dedicated condition dimension containing a one-hot encoded target class vector spatially broadcast across all cells.

These regeneration effects only emerge reliably when *sample pooling* is enabled during training. Without pooling, models fail to recover from perturbations and instead drift or collapse (see appendix A.4). All results reported in this section therefore use sample pooling by default.

Regeneration quality consistently improves as the perception module becomes more expressive, either through larger receptive fields or learnable spatial adaptivity. The Combined perception achieves both the lowest steady-state LPIPS and the highest regeneration scores (98.32% for damage, 99.64% for mutation). In contrast, limited or fixed perceptions such as Sobel filtering show higher reconstruction error and significantly weaker recovery (84.92%). This suggests that regeneration is correlated with the network's ability to integrate information beyond strictly local neighborhoods. The Attention 3x3 perception achieves excellent generation quality (LPIPS 0.009 at Stage 1), outperforming standard Conv 3x3 despite having the same $3 \times 3$ receptive field size. However, we observed that training proved substantially less stable than convolutional approaches, requiring careful hyperparameter tuning to avoid collapse.

The LPIPS curves in fig. 7a illustrate successful adaptation when the global target condition is switched mid-evolution. Figure 7b provides a qualitative comparison across perception types, showing that richer perception results in both faster and more complete regrowth.

Table 1: Emoji generation performance: LPIPS loss and regeneration percentage after damage and mutation for different perception modules. For NCA-based models, we report LPIPS at three rollout stages: Stage 1 (within the training horizon, 64–96 update steps), Stage 2 (after additional rollout, right before applying damage or mutation, 149 steps), and Stage 3 (after the regeneration phase, 449 steps). U-Net and GAN are single-step models without iterative updates; their scores are evaluated at the same three pipeline stages but do not involve further rollouts. Lower LPIPS is better; higher Regeneration is better.

| | Damage | | | | Mutation | | | | Params. |
|---|---|---|---|---|---|---|---|---|---|
| Perception | LPIPS ↓ Stage 1 (64–96) | LPIPS ↓ Stage 2 (149) | LPIPS ↓ Stage 3 (449) | Reg. ↑ (%) | LPIPS ↓ Stage 1 (64–96) | LPIPS ↓ Stage 2 (149) | LPIPS ↓ Stage 3 (449) | Reg. ↑ (%) | |
| Sobel | 0.152 | 0.124 | 0.144 | 84.92 | 0.151 | 0.124 | 0.138 | 93.64 | 8k |
| Conv 3x3 | 0.065 | 0.060 | 0.128 | 58.74 | 0.068 | 0.060 | 0.070 | 95.75 | 23k |
| U-Net | - | 0.070 | 0.064 | 100 | - | 0.061 | 0.055 | 99.26 | 25k |
| GAN | - | 0.042 | 0.037 | 100 | - | 0.042 | 0.039 | 100 | 25k |
| Deformable 3x3 | 0.040 | 0.035 | 0.041 | 96.17 | 0.040 | 0.035 | 0.039 | 98.33 | 27k |
| U-Net | - | 0.028 | 0.026 | 100 | - | 0.029 | 0.026 | 100 | 44k |
| GAN | - | 0.018 | 0.017 | 100 | - | 0.018 | 0.018 | 100 | 44k |
| Conv 5x5 | 0.032 | 0.025 | 0.042 | 89.58 | 0.032 | 0.025 | 0.029 | 98.34 | 51k |
| U-Net | - | 0.024 | 0.024 | 100 | - | 0.022 | 0.024 | 100 | 69k |
| GAN | - | 0.015 | 0.015 | 100 | - | 0.017 | 0.017 | 100 | 69k |
| Comb. 3x3+5x5D | **0.003** | **0.001** | **0.005** | **98.32** | **0.003** | **0.001** | **0.003** | **99.64** | 71k |
| Attention 3x3 | 0.009 | 0.010 | 0.026 | 85.78 | 0.009 | 0.010 | 0.012 | 95.05 | 80k |

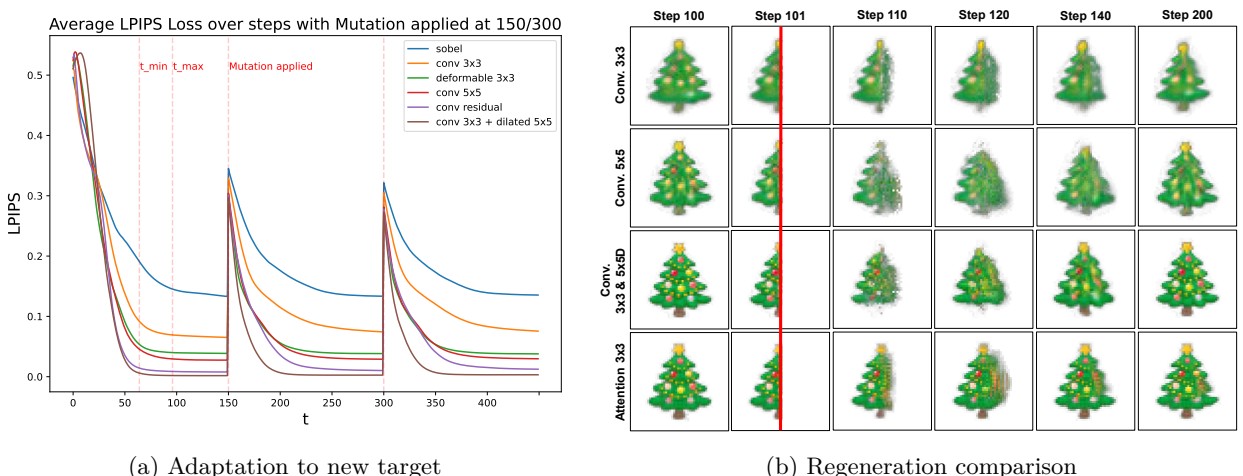

(a) Adaptation to new target

(b) Regeneration comparison

Figure 7: Emoji generation dynamics illustrating adaptation and regeneration robustness. (a) Example of NCA adaptation: the state transition sequence when the target emoji condition is switched mid-evolution. (b) Comparison of regeneration quality after 50% vertical damage for three perception strategies (rows, top to bottom: standard 3×3 Conv, standard 5×5 Conv, 3×3 Conv + 5×5 Dilated Conv, and 3x3 attention perception
). Each row shows the state before damage, immediately after damage, and during subsequent recovery. Consistent with the quantitative results in Table 1, the more complex perception achieves near-complete visual regeneration, while the 3×3 Conv shows limited recovery.

### 3.1.3 Edge-to-Handbag (E2H) Task

**Prior Work.** The integration of NCAs with adversarial training was introduced by (Otte et al., 2021b), who demonstrated that NCAs could function as generators within a GAN framework for image synthesis. Their work showed that combining the self-organizing properties of NCAs with adversarial objectives enables the generation of more realistic and diverse images compared to MSE-based training alone. The Edge-to-Handbag (E2H) dataset (Isola et al., 2018) provides a challenging conditional image-to-image translation task, where the model must translate sparse edge maps into detailed, textured handbag images. This requires both handling complex conditional inputs and generating intricate visual details—capabilities that benefit from adversarial training objectives.

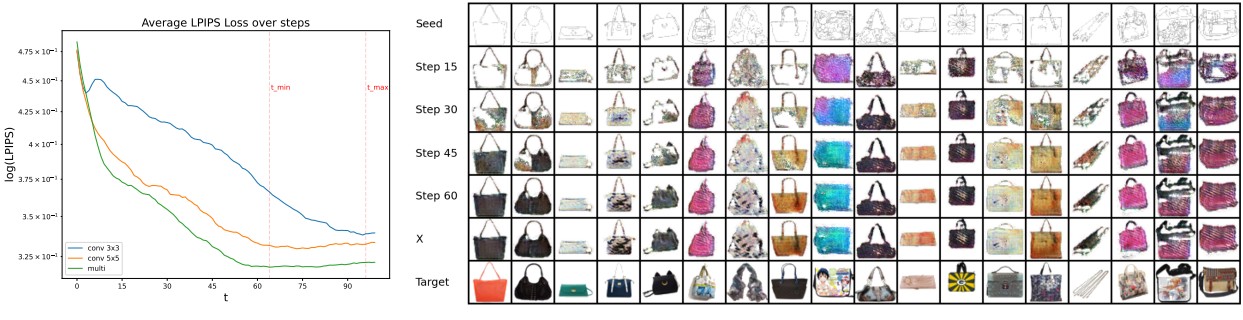

(a) LPIPS loss over generation          (b) Generation examples over time

Figure 8: Edge-to-Handbag (E2H) conditional generation results using NCAs under adversarial training. (a) Learned Perceptual Image Patch Similarity (LPIPS) loss during generation for different perception modules (standard 3x3 Conv, wider 5x5 Conv, and Combined [3x3 Conv + 5x5 Dilated Conv D=2]). Lower LPIPS indicates better perceptual quality. (b) Visualization of the generation sequence from initial seed state to final generated handbag images for multiple samples. Last column shows the ground truth target image corresponding to the input edges.

**Our Extension.** We apply adversarial training to the E2H task within our modular framework, systematically comparing different perception architectures under this training regime. The NCA rollout follows algorithm 1, but replaces the reconstruction loss with a Wasserstein GAN objective (Arjovsky et al., 2017) with Gradient Penalty (WGAN-GP) (Gulrajani et al., 2017): the NCA acts as the generator producing RGB images ($C_v = 3$) conditioned on the input edge map, while a critic network is trained concurrently to distinguish real from generated samples.

**Results.** As illustrated in fig. 8, perceptions with broader spatial context yielded superior results even under this adversarial training regime. The loss curves in fig. 8a, plotting LPIPS (Zhang et al., 2018) over time, clearly show that perceptions with larger receptive fields (Conv 5x5 and Combined) resulted in lower LPIPS loss compared to the standard Conv 3x3, correlating with higher perceptual quality in the generated images. Qualitative examples of the generation process are shown in fig. 8b, visualizing the evolution from the initial seed state towards the final handbag image across multiple samples. These results reinforce our findings from other tasks: equipping NCAs with perception modules that capture more global information significantly enhances their performance on complex generation tasks, regardless of whether the objective is direct reconstruction or adversarial training.

### 3.1.4 Latent Space NCAs

**Prior Work.** Recent work by Menta et al. (2024) introduced latent space NCAs for image restoration and inpainting tasks, demonstrating that NCAs can effectively operate in the compressed representations learned by autoencoders. This methodology builds upon principles common in state-of-the-art generative models such as Stable Diffusion (Rombach et al., 2022), which leverage learned latent representations from models like VQVAE (van den Oord et al., 2018) to enable efficient high-resolution generation.

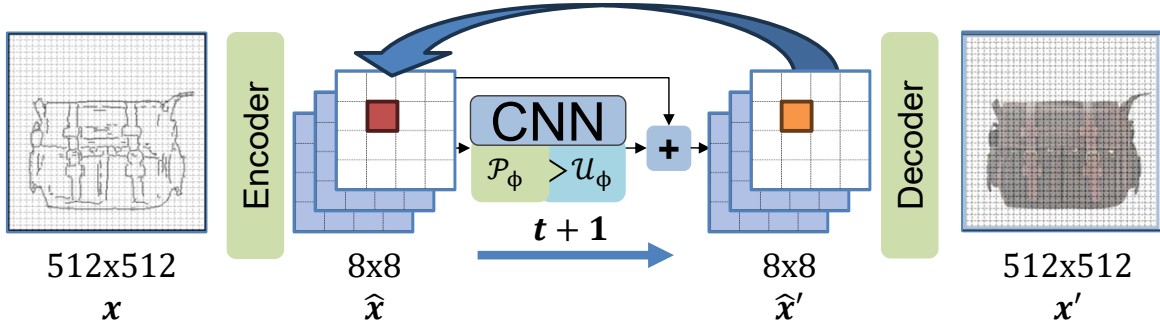

Figure 9: Latent space NCA architecture for high-resolution image generation and inpainting. A pre-trained VQVAE encoder compresses the input/seed state into a compact latent representation. The NCA evolves within this learned latent space through iterative local updates.

**Our Extension.** We extend latent space NCAs in two directions. The training procedure adapts algorithm 1 by wrapping the NCA rollout with a pre-trained encoder and decoder; the full latent training algorithm is detailed in algorithm 2. First, we apply latent NCAs to *conditional multi-class generation* using both the Emoji and E2H datasets, enabling generation at higher resolutions ($512\times512$ for Emoji, $256\times256$ for E2H). A pre-trained VQVAE encodes the seed state and target images into compressed latent representations, allowing the NCA to evolve in this lower-dimensional space. The evolved latent state is decoded back to pixel space, and the loss $\mathcal{L}$ (MSE) is computed in the pixel domain to train the NCA. Second, we explore latent NCAs for *high-resolution image inpainting* on CelebA (Liu et al., 2015) using a two-stage protocol: first training a VQVAE on both perturbed (with random circular patches removed) and complete images to learn a robust latent manifold, then training the NCA entirely in latent space to evolve masked latent encodings toward their complete counterparts through context-aware regeneration.

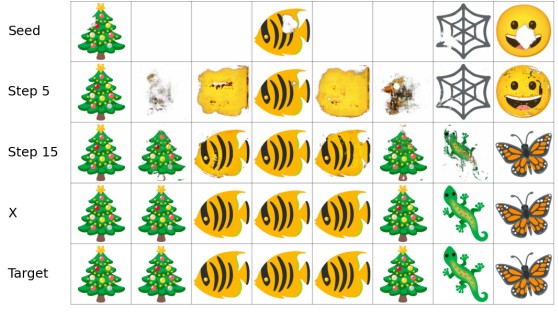

(a) Emoji generation at $512\times512$

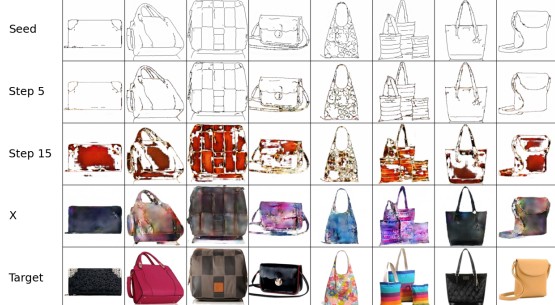

(b) E2H generation at $256\times256$

Figure 10: Latent space NCA generation for conditional image synthesis. The NCA evolves in the compressed latent space of a pre-trained VQVAE, enabling efficient high-resolution generation for both (a) multi-class emoji generation and (b) edge-to-handbag translation.

**Results.** For conditional generation (fig. 10), latent space NCAs successfully synthesized high-resolution images for both Emoji and E2H tasks, achieving faster growth with substantially less computation compared to pixel-space models due to their compact representation. Unlike traditional pixel-space cellular automata, which require simulation steps proportional to image dimensions, latent space evolution enables efficient scaling to higher resolutions. For the CelebA inpainting task (fig. 11), the latent NCA demonstrated effective context-aware regeneration, progressively reconstructing missing facial features over evolution steps by leveraging local communication in the learned latent manifold. These results demonstrate that latent space NCAs can successfully handle both conditional generation and restoration tasks at higher resolutions while maintaining computational efficiency.

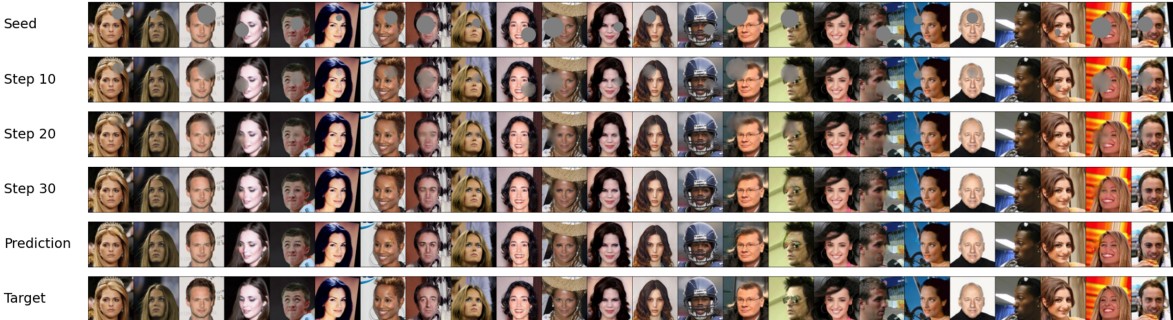

Figure 11: Latent NCA regeneration on CelebA inpainting. The sequence shows the evolution of the latent state decoded to pixel space at intermediate steps. Starting from the masked image's latent representation (top), the NCA progressively regenerates missing facial features through local communication in latent space.

### 3.1.5 Scalability Analysis: Latent vs. Pixel-Space NCAs

While pixel-space NCAs excel at small-scale generation tasks, they face a fundamental scalability barrier: the state size grows quadratically with image resolution. We demonstrate that latent-space NCAs are not merely an optimization, but a necessary architectural choice for generating high-dimensional images. To quantify this limitation, we benchmark latent-space NCAs (utilizing a VQVAE encoder/decoder with $8\times$ spatial compression) against pixel-space NCAs on an RTX 4090 (24GB VRAM). For all experiments, the number of NCA steps is set to half the grid width ($T = W/2$). Because the NCA grows from a single central seed and information propagates at a maximum rate of one cell per step, this represents the theoretical minimum number of steps required for the signal to propagate from the center to the grid boundaries.

| Setup | Grid | NCA Steps | Eval (ms) | Eval Mem (MB) | Train Mem (MB) |
|---|---|---|---|---|---|
| **Latent Space** | | | | | |
| $256\times256$ | $32\times32$ | 16 | 0.33 | 357 | 474 |
| $512\times512$ | $64\times64$ | 32 | 0.36 | 561 | 1461 |
| $1024\times1024$ | $128\times128$ | 64 | 0.50 | 1514 | 7359 |
| **Pixel Space** | | | | | |
| $256\times256$ | $256\times256$ | 128 | 0.38 | 398 | 14254 |
| $512\times512$ | $512\times512$ | ——————*Out of Memory*—————— | | | |
| $1024\times1024$ | $1024\times1024$ | ——————*Out of Memory*—————— | | | |

Table 2: Latent vs. pixel-space NCA scalability (batch size 1). Latent models achieve $38\times$ lower training memory at $256 \times 256$ and scale to $1024 \times 1024$, while pixel-space NCAs become infeasible beyond $256 \times 256$ on a 24GB GPU.

As demonstrated in table 2, the memory requirements for pixel-space NCAs scale quadratically with image resolution, creating a fundamental limitation for high-resolution generation. At $256 \times 256$ resolution, pixel-space NCAs already consume 14.3 GB of training memory per sample. Attempting $512 \times 512$ generation results in out-of-memory errors on consumer hardware with 24 GB VRAM, primarily because the grid size requires increasing the step count to 256 to allow full propagation, compounding the gradient memory cost. In contrast, latent-space NCAs operating on the compressed $32 \times 32$ representation achieve a $38\times$ reduction in training memory at $256 \times 256$ (474 MB). Critically, the reduced grid size allows for proportionally fewer steps (e.g., only 64 steps for a $1024 \times 1024$ image with a $128 \times 128$ latent grid), enabling generation at resolutions that are completely infeasible in pixel space. These results establish that latent-space evolution is a necessary architectural choice for scaling NCAs to practical high-resolution image generation.

### 3.2 Texture Generation

**Prior Work.** Niklasson et al. (2021) demonstrated that NCAs can learn to generate and self-organize complex textures by training on style-based loss functions. Their approach used VGG-based Gram matrix losses to capture texture statistics from the Oxford Describable Textures Dataset (DTD) (Cimpoi et al., 2014), enabling NCAs to produce dynamic, temporally consistent textures that emerge from fixed initial states. This work established that NCAs could move beyond simple pattern replication to synthesize rich textural patterns through self-organization, with the texture characteristics encoded in the learned update rules rather than explicitly in the initial state.

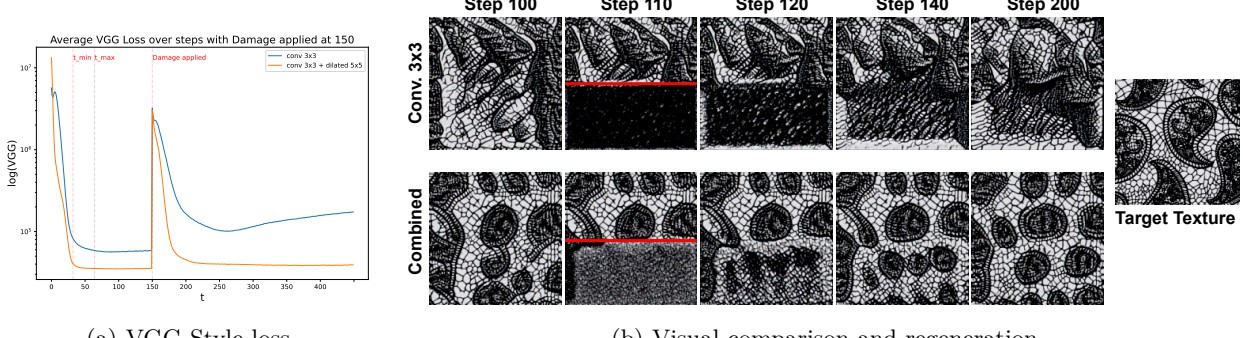

(a) VGG Style loss

(b) Visual comparison and regeneration

Figure 12: NCA texture generation and regeneration dynamics for the 'lacelike' DTD texture. (a) VGG Style loss progression over evolution steps, showing recovery after 50% horizontal removal at t=150. (b) Visual comparison including the target texture and generated results. It illustrates the state before perturbation, immediately after, and the final regenerated states for models using standard ('Conv 3x3') versus enhanced ('Combined' [MultiPerception: 3x3 Conv + 5x5 Dilated D=2]) perception. Note how the Combined perception more accurately reproduces both fine local details and larger structural patterns compared to the simpler Conv 3x3 perception, both in the initial generation and during regeneration.

**Our Extension.** We apply the texture synthesis approach following the training procedure in algorithm 1 to systematically evaluate how different perception modules affect both texture generation quality and regeneration robustness. We train NCAs on a 168×168 grid using the DTD dataset. Unlike the image generation tasks that use pixel-wise reconstruction losses, the texture task uses a composite loss $\mathcal{L} = \mathcal{L}_{\text{style}} + \lambda \mathcal{L}_{\text{overflow}}$ where $\mathcal{L}_{\text{style}}$ is a VGG-based sliced optimal transport loss over VGG layers $l \in \mathcal{L}_{\text{VGG}}$ and $P$ random unit-norm projections $\mathbf{w}_p$:

$$\mathcal{L}_{\text{style}} = \sum_{l,\,p} \left\| \text{sort}\left(\mathbf{w}_p^\top \phi_l(\hat{\mathbf{x}})\right) - \text{sort}\left(\mathbf{w}_p^\top \phi_l(\mathbf{x}_{\text{target}})\right) \right\|_2^2, \tag{5}$$

and $\mathcal{L}_{\text{overflow}}$ is a regularization term penalizing activations outside their valid ranges:

$$\mathcal{L}_{\text{overflow}} = \text{mean}(\text{ReLU}(\hat{\mathbf{x}}_{\text{rgb}} - 1) + \text{ReLU}(-\hat{\mathbf{x}}_{\text{rgb}})) + \text{mean}(\text{ReLU}(|\hat{\mathbf{x}}_{\text{hidden}}| - 1)), \tag{6}$$

penalizing RGB values outside $[0, 1]$ and hidden channels outside $[-1, 1]$. Training employs the Sample Pool mechanism (section 2.4) with evolution windows of 32-54 steps per iteration to improve stability. We compare two perception strategies: standard $3\times3$ convolutional perception and an enhanced 'Combined' strategy using MultiPerception ($3\times3$ Conv + $5\times5$ Dilated Conv D=2) to provide a broader receptive field. Critically, we assess not just generation quality but also robustness through regeneration experiments, applying significant perturbations (50% horizontal or vertical removal) after 150 evolution steps to test self-repair capabilities.

**Results.** The enhanced 'Combined' perception strategy substantially improved the visual quality of generated textures, particularly evident in patterns requiring broader spatial context such as the 'lacelike' texture shown in fig. 12b. This texture was chosen for detailed analysis as its structure contains both fine-grained local details and complex larger arrangements, providing a suitable test case for evaluating perception capabilities. Quantitative evaluation using the regeneration metric (eq. (3)) confirmed the visual superiority: the Combined perception achieved significantly higher recovery (99.88%) compared to standard $3\times3$ convolution

(96.38%). As shown in fig. 12a, the VGG style loss progression demonstrates that the Combined perception recovers more effectively after damage. These results suggest that broader receptive fields contribute to both better initial texture generation and more effective self-repair capabilities in texture synthesis tasks.

### 3.3 Classification (MNIST, CIFAR-10)

**Prior Work.** Randazzo et al. (2020) introduced self-classifying NCAs for MNIST digit recognition, demonstrating that NCAs could perform classification by evolving class predictions over time through local interactions. In their approach, the NCA states include fixed input channels containing the image and evolving classification channels (one for each class). Through iterative updates based on pixel-wise losses (MSE or Cross-Entropy), these classification channels converge to represent the correct class, with the final prediction derived by spatially averaging the class channels and selecting the maximum as visualized in fig. 13.

**Our Extension.** The classification task adapts algorithm 1 with two key differences: (i) the input image is placed into *fixed* (non-evolving) channels, and (ii) the loss is computed on dedicated classification channels instead of visible channels. Specifically, the state includes $K$ evolving classification channels (one per class), and the loss $\mathcal{L}$ is either pixel-wise MSE or Cross-Entropy between these channels $\mathbf{c}_T \in \mathbb{R}^{H \times W \times K}$ and a spatially broadcast one-hot target $\mathbf{y}_{\mathrm{oh}} \in \{0, 1\}^{H \times W \times K}$:

$$\mathcal{L}_{\mathrm{MSE}} = \frac{1}{HWK} \sum_{i,j,k} \left( \mathbf{c}_T[i,j,k] - \mathbf{y}_{\mathrm{oh}}[i,j,k] \right)^2, \tag{7}$$

$$\mathcal{L}_{\mathrm{CE}} = -\frac{1}{HW} \sum_{i,j} \log \frac{\exp(\mathbf{c}_T[i,j,y_{i,j}^*])}{\sum_{k'=1}^{K} \exp(\mathbf{c}_T[i,j,k'])}, \quad y_{i,j}^* = \arg\max_k \mathbf{y}_{\mathrm{oh}}[i,j,k]. \tag{8}$$

The final class prediction is obtained by spatially averaging each classification channel and selecting the argmax: $\hat{y} = \arg\max_k \frac{1}{HW} \sum_{i,j} \mathbf{c}_T[i,j,k]$. The full classification training algorithm is detailed in algorithm 3.

While we validate the MNIST self-classification approach within our framework, our key contribution is extending the self-classification paradigm to the significantly more challenging *CIFAR-10 dataset*. CIFAR-10 presents much higher visual complexity (airplanes, trucks, animals, etc.) with color images and greater intra-class variability compared to grayscale MNIST digits. For CIFAR-10, we systematically evaluate how different architectural choices affect classification performance: (i) standard 3×3 convolutional perception, (ii) 3×3 convolution with Sample Pooling (SP) for long-term stability, and (iii) an enhanced perception with residual connections and a wider receptive field (3×3 Conv + 5×5 Dilated Conv D=2).

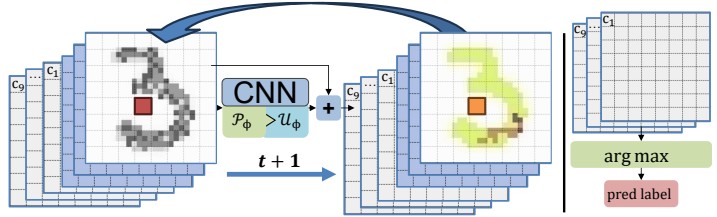

Figure 13: Self-classifying NCA architecture. The cell state is partitioned into fixed input channels (containing the image to classify), hidden channels for computation, and evolving classification channels (one per class). Through iterative local updates, classification values propagate and converge across all spatial locations. The final prediction is obtained by spatially averaging each class channel and selecting the maximum.

#### 3.3.1 MNIST Self-Classification

**Validation Results.** We first validate the self-classification approach on MNIST using a standard 3×3 convolutional perception module. We compare MSE and Pixel-wise Cross-Entropy (PCE) losses applied to these classification channels, with targets being one-hot encoded spatial representations (target=1 in the true class channel at all locations, 0 otherwise).

Table 3 summarizes the classification accuracy, showing high performance during stable evolution (e.g., steps 20-30) for both MSE (94.6%) and PCE (95.3%) losses. To test adaptation, the input digit (held in

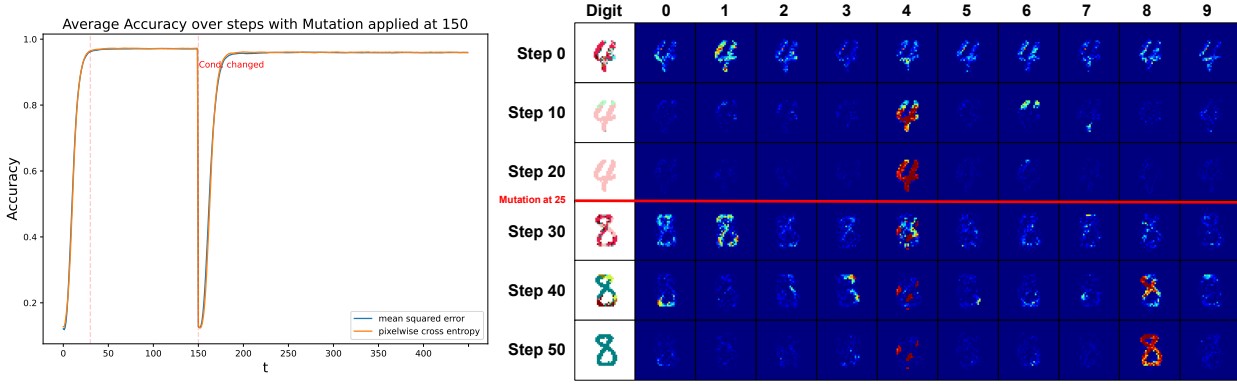

| (a) Per-pixel accuracy over time | (b) Evolving class channels adapting to input mutation |

Figure 14: MNIST self-classification and adaptation dynamics. (a) Per-pixel classification accuracy over evolution steps, showing the effect of mutating the input digit at t=150. (b) Visualization showing the input digit (fixed) and the state of the 10 evolving classification channels before and after the input mutation.

the fixed channel) was mutated to a different digit after 150 steps. The table shows accuracy just before mutation (step 149) and long after (step 449), demonstrating the network's ability to dynamically adapt its pixel-wise predictions to the new input digit. The "Regeneration" column quantifies this adaptation capability by measuring the recovery of per-pixel accuracy relative to the pre-mutation level after the input change; both loss functions enable strong adaptation (99.31% for MSE, 98.46% for PCE). Figure 14 provides further illustration: fig. 14a plots the per-pixel accuracy over time, including the dip and recovery following the mutation event at t=150, while fig. 14b shows a visual example of the evolving classification channels adapting to the changed input digit.

Table 3: Mean Per-Pixel Accuracy and Adaptation Recovery on MNIST. For NCAs, columns show average accuracy during stable evolution (20-30 steps), just before input mutation (step 149), long after mutation (step 449), and the calculated adaptation recovery (%). GAN is a single-step model evaluated only at the equivalent final timestep.

| Loss | Model | Acc. (20–30) ↑ | Acc. (149) ↑ | Acc. (449) ↑ | Reg. (%) ↑ | Params. |
|------|-------|----------------|--------------|--------------|------------|---------|
| MSE  | NCA   | 94.6%          | 97.0%        | 96.4%        | 99.31      | 23k     |
| PCE  | NCA   | 95.3%          | 97.8%        | 96.4%        | 98.46      | 23k     |
| PCE  | GAN   |                | 94.78%       |              | N/A        | 25k     |

We note that MNIST digit classification is a well-solved problem, with state-of-the-art convolutional neural networks routinely achieving near-human performance (Cireşan et al., 2012). Our NCA implementation serves as a proof-of-concept demonstration of the self-classifying paradigm rather than a competitive method. For comparison to the image-to-image NCA approach, we trained a GAN baseline with comparable model size (25k parameters) achieving 94.78% accuracy. However, larger and more sophisticated architectures have no difficulty approaching near-perfect classification performance. The key contribution here is showing that NCAs can learn to perform classification through iterative local updates, not that they outperform established methods for this task.

### 3.3.2 CIFAR-10 Classification

Extending to CIFAR-10 addresses a significantly higher visual complexity with color images. The NCA state includes the fixed 3-channel RGB input, hidden channels, and 10 evolving classification channels trained with MSE loss on one-hot spatial targets. We evaluate three configurations: (i) standard 3×3 convolution, (ii) 3×3 with Sample Pooling (SP), and (iii) enhanced perception (Residual + 5×5 Dilated Conv D=2).

Table 4: Mean Image Classification Accuracy on CIFAR-10. Columns show average accuracy during the training window (32-64 steps) and accuracy at a later timestep (step 300).

| Perception Strategy | Acc. (32-64) | Acc. (300) |
|---|---|---|
| Conv 3x3 | 43.83% | 8.14% |
| Conv 3x3 + SP | 41.11% | 46.13% |
| Residual + Dilated 5x5 Conv | 72.98% | 54.37% |

As shown in table 4 and fig. 15, the enhanced perception dramatically improved performance, achieving 72.98% accuracy (steps 32-64) versus 43.83% for standard 3×3 convolution. Sample Pooling proved critical for long-term stability: without SP, accuracy collapsed to 8.14% by step 300, while with SP, it maintained 46.13%. The enhanced perception also showed better stability (54.37% at step 300). These results demonstrate that self-classifying NCAs successfully extend to complex natural image datasets beyond MNIST, with architectural improvements (wider receptive fields, residual connections) and training mechanisms (SP) proving equally beneficial for classification as well as for generation tasks.

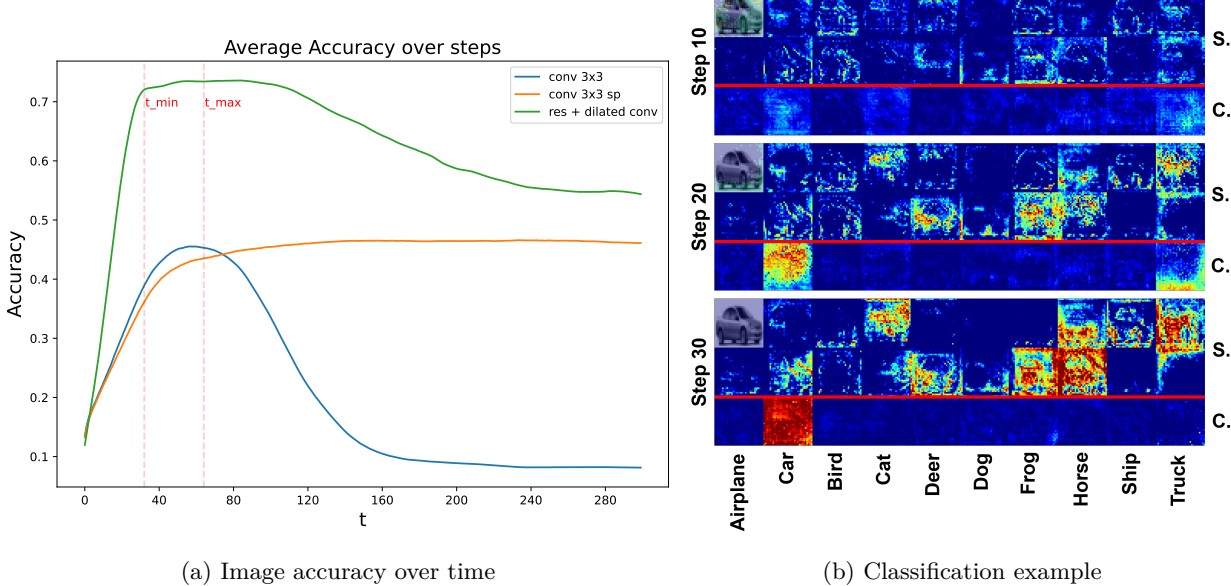

(a) Image accuracy over time

(b) Classification example

Figure 15: CIFAR-10 classification performance. (a) Image classification accuracy over evolution steps for three perception strategies: Conv 3x3, Conv 3x3 with Sample Pooling (SP), and Residual + Dilated 5x5 Conv. (b) Visual examples showing the input RGB image, hidden channels, and classification channels (bottom row) for one sample.

Similar to MNIST, CIFAR-10 classification is well-established, with state-of-the-art deep residual networks achieving accuracies up to 99.5% (Han et al., 2017; Dosovitskiy et al., 2021). Our NCA implementation serves as a proof-of-concept that the self-classifying paradigm can extend to more complex natural image datasets beyond simple digits. The contribution is demonstrating that NCAs can learn to classify natural images through iterative local updates, rather than competing with optimized architectures designed specifically for this benchmark.

### 3.4 Video Transition (MovingMNIST)

**Prior Work.** While NCAs have been successfully applied to static pattern generation, texture synthesis, and classification, their application to temporal sequence prediction and video dynamics has remained relatively unexplored. Video prediction tasks require learning not just spatial patterns but also temporal transition rules that govern how patterns evolve over time. The MovingMNIST dataset (Srivastava et al., 2016) provides

a controlled testbed for video prediction, where digits move with constant velocity and bounce off boundaries, requiring models to learn both spatial and temporal coherence.

**Our Extension.** We introduce a novel application of NCAs to video frame prediction, adapting algorithm 1 to a state-to-state transition problem. The NCA's grid state is initialized with a sequence of $k$ consecutive frames from a MovingMNIST video, encoded into the first $k$ visible channels ($C_v = k$) of the state tensor, while the remaining hidden channels are initialized to zero. The NCA evolves this initial state over multiple steps, with the training objective being to match the next sequence of $k$ frames (i.e., the original sequence shifted forward by one timestep) using MSE loss. Critically, we employ the Sample Pool mechanism (appendix A.4), where the model's own predictions are recycled as initial states for subsequent training iterations, encouraging the learning of self-stabilizing and temporally coherent dynamics.

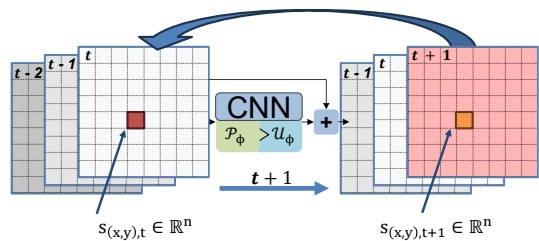

Figure 16: Video prediction NCA architecture for MovingMNIST. The cell state is initialized with $k$ consecutive video frames encoded in the first $k$ channels, with remaining hidden channels set to zero. The NCA evolves this temporal state representation over multiple steps to predict the next $k$ frames (the input sequence shifted forward by one timestep).

**Results.** As shown in fig. 17, NCAs successfully learn temporal transition dynamics for video prediction. The MSE loss curves (fig. 17a) demonstrate that the NCA state converges as it evolves from input frames toward target frames during each transition step. Visual examples (fig. 17b) illustrate the iterative prediction process across the frame sequence. These results establish video prediction as a possible application domain for NCAs, extending their utility beyond static pattern generation to temporal sequence modeling.

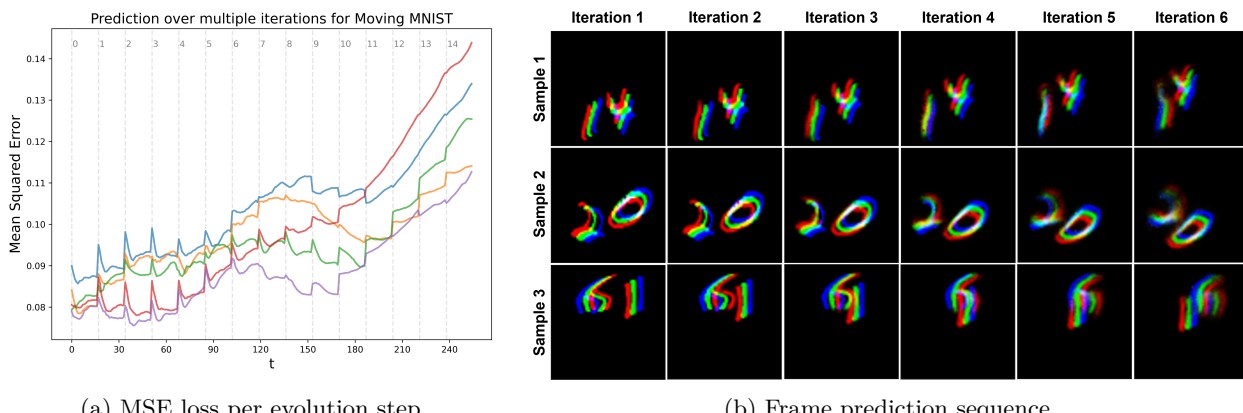

(a) MSE loss per evolution step

(b) Frame prediction sequence

Figure 17: MovingMNIST video prediction with NCAs. (a) MSE loss over NCA evolution steps for each frame-to-frame transition, showing how the NCA state evolves from the initial frames (state $t$) toward the target frames (state $t + n$). (b) Visual examples of the prediction sequence: starting from input frames, the NCA iteratively evolves its state to predict the next frames in the video, demonstrating learned temporal transition dynamics.

## 4    Discussion

In the previous sections, we have presented a systematic analysis of the current research on *Neural Cellular Automata* using our proposed framework **NCAtorch**. The modular design of **NCAtorch** and the technical consolidation and engineering contributions allowed an in-depth study of NCAs in a wide range of computer-vision and image generation applications.

While our results show the great potential of learnable update rules in *Cellular Automata* and their application in vision problems, the current research of this novel neural architecture is still in its basic phase. Starting from these basic findings, three main research directions appear to be promising for the near future:

**I Utilization of the rich theoretical properties of CA:** as discussed in section 1, there is an extensive body of literature showing intriguing theoretical properties of *Cellular Automata*. From computational properties like *Turing Completeness* to *Wolfram's* core idea to model complex systems which exceed the capabilities or computational complexity of differential equations. Transferring these into data driven, learning applications has the potential to be the key benefit of the novel architecture type.

**II Scaling of NCAs to handle state-of-the-art vision problems:** While the experiments conducted in this paper are able to show the potential of NCAs, they are still far from practically impacting current learning and vision applications. Scaling NCAs to tackle state-of-the-art computer-vision, image generation or physical modeling applications is therefore paramount for an acceptance of this new methodology. Our experiments with latent space NCAs in section 3.1.4 have shown that this is possible. The next step would be to scale this approach on large datasets, requiring a lot of compute.

**III) Investigation of the close theoretical relationship of NCAs with other neural architecture types:** a closer look at the NCA architecture shows obvious theoretical relations to other, well established neural architectures. It does not take a lot of imagination to see NCAs as special or more generalized versions of RNNs, Graph Neural Networks, Diffusion Models or Normalizing Flows. This raises interesting followup questions: can we either transfer the strong theoretical properties of CA to other network types, or could we even formulate a general unified architecture?

In any case, we will continuously develop **NCAtorch** to provide an easy-to-use and extend toolbox for all of these possible future research avenues.

## Broader Impact Statement

This work presents a systematic review and open-source reference implementation of Neural Cellular Automata (NCA). By consolidating existing NCA methods into a unified, accessible framework, we aim to lower the barrier to entry for researchers across disciplines, including computer vision, biological modeling, and physical simulation, who may benefit from the self-organizing and local-computation properties of NCAs.

The decentralized, local-interaction nature of NCAs makes them a compelling candidate for computationally efficient and interpretable models, which could have positive implications for resource-constrained applications. Furthermore, the theoretical connections between NCAs and biological systems may offer new tools for researchers studying morphogenesis, pattern formation, and complex systems.

We do not foresee direct negative societal consequences specific to this work. The generative capabilities demonstrated here operate at a scale and fidelity well below state-of-the-art image synthesis models, and the primary contribution is a research tool rather than a deployable system. As with any generative modeling framework, however, future extensions toward higher-fidelity synthesis should be accompanied by appropriate consideration of potential misuse.

## Funding Acknowledgement

The authors acknowledge the financial support by the German Federal Ministry of Research, Technology and Space (BMFTR) in the program "Forschung an Fachhochschulen in Kooperation mit Unternehmen (FH-Kooperativ)" within the joint project "KI-Bohrer" under grant 13FH525KX1.

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

# A  Appendix

## A.1  Additional Related Work

This appendix expands the discussion of related research by providing a broader survey of work on NCAs. In contrast to the related work discussed in the main paper, these methods have not yet been implemented and evaluated in our framework.

**Self-organizing pattern formation.**  Early work established that NCAs can learn robust, self-organizing update rules for growth and regeneration (Mordvintsev et al., 2020; Randazzo et al., 2020; Niklasson et al., 2021). Subsequent models show that NCAs can learn spatio-temporal reaction–diffusion dynamics and reproduce Turing-like biological pattern formation (Richardson et al., 2024).

**Memory and morphogenetic control.**  A recent line of work introduces hidden internal channels that let each cell store, modify, and propagate information locally, enabling a single seed to generate multiple distinct morphologies without a global controller (Guichard et al., 2025).

**Generative and diffusion-based models.**  NCAs have been adapted for high-resolution image generation, including architectures that combine NCA dynamics with diffusion processes (Kalkhof et al., 2024b; Mittal et al., 2025), and growing-from-seed image models (Elbatel et al., 2024).

**Physical simulation and scientific modeling.**  NCAs can emulate complex physical processes while remaining computationally lightweight; for example, learned NCAs model metal solidification and capture grain growth behaviour with extrapolation beyond the training regime (Tang et al., 2023).

**Embodied systems and AI reasoning.**  NCAs have been applied to tasks beyond continuous pattern formation, including controlling modular self-assembling robots and solving abstract reasoning tasks such as ARC-AGI via emergent rule learning (Hartl et al., 2025).

## A.2  Latent Space NCA Training Algorithm

The latent space NCA training procedure (algorithm 2) extends the pixel-space algorithm (algorithm 1) by wrapping the NCA rollout with a pre-trained encoder-decoder pair. A frozen VQVAE encoder $\mathcal{E}$ compresses the seed and target into a compact latent grid, the NCA evolves entirely in this lower-dimensional space, and the frozen decoder $\mathcal{D}$ maps the final latent state back to pixel space for loss computation. Crucially, the encoder and decoder weights are fixed; only the NCA parameters $\phi$ are updated. The sample pool operates on latent states rather than pixel images.

## A.3  Self-Classification NCA Training Algorithm

The classification variant (algorithm 3) adapts the generative training procedure (algorithm 1) for recognition tasks. Two key differences distinguish it from the generative setting: (i) the input image $\mathbf{x}_{\text{input}}$ is placed into *fixed* (non-evolving) channels rather than acting as a seed to be overwritten, and (ii) the loss is computed on $K$ dedicated *evolving* classification channels instead of visible output channels. The target is a one-hot vector $\mathbf{y} \in \{0,1\}^K$ spatially broadcast to all cell positions, and the final class prediction is obtained by spatially averaging each classification channel and selecting the argmax.

## A.4  Impact of Sample Pooling

To empirically validate the critical role of Sample Pooling in our framework, we conducted an ablation study comparing training with and without SP using a standard 3x3 convolutional perception module. As detailed in table 5 and visualized in fig. 18, the inclusion of SP drastically improves regeneration capabilities following perturbations. Without SP, the system failed to recover effectively, resulting in large negative regeneration scores (-196.18% for damage, -125.05% for mutation), indicative of significant degradation post-perturbation.

---

**Algorithm 2:** Latent Space NCA Training

---

**Input** : Training sample $(\mathbf{x}_{\text{seed}}, \mathbf{c}, \mathbf{x}_{\text{target}})$, NCA model $\mathcal{F}_\phi$,
pre-trained encoder $\mathcal{E}$ and decoder $\mathcal{D}$ (frozen),
loss $\mathcal{L}$, step range $[T_{\min}, T_{\max}]$, fire rate $p_{\text{fire}}$

**Output:** Trained NCA parameters $\phi^*$

**1** Initialize optimizer, LR scheduler, sample pool $\mathcal{P}_{\text{pool}} \leftarrow \emptyset$;

**2 for** *each training iteration* **do**

**3**      Draw $(\mathbf{x}_{\text{seed}}, \mathbf{c}, \mathbf{x}_{\text{target}})$ from dataset $\mathcal{D}$;

**4**      Encode seed to latent space: $\mathbf{z}_0 \leftarrow \mathcal{E}(\mathbf{x}_{\text{seed}})$ ;           `// frozen encoder, no gradient`

     `// Sample Pool:  optionally replace z₀ with a previously evolved latent state`

**5**      **if** $|\mathcal{P}_{\text{pool}}| > 0$ **then**

**6**          With probability $r_{pool}$: replace $\mathbf{z}_0$ with a random sample from $\mathcal{P}_{\text{pool}}$;

**7**          Optionally apply spatial damage or condition mutation;

     `// Multi-step NCA rollout in latent space`

**8**      Sample rollout length $T \sim \mathcal{U}[T_{\min}, T_{\max}]$;

**9**      **for** $t = 0$ to $T - 1$ **do**

**10**          $\mathbf{p}_t \leftarrow \mathcal{P}_\phi(\mathbf{z}_t)$ ;           `// perception on latent grid`

**11**          $\Delta\mathbf{z}_t \leftarrow \mathcal{U}_\phi(\mathbf{p}_t)$ ;          `// update:  compute latent increment`

**12**          Sample mask $\mathbf{m}_t \sim \text{Bernoulli}(p_{\text{fire}})^{H' \times W'}$ ;      `// H'×W' is the latent grid size`

**13**          $\mathbf{z}_{t+1} \leftarrow \mathbf{z}_t + \mathbf{m}_t \odot \Delta\mathbf{z}_t$;

     `// Decode to pixel space and compute loss`

**14**      $\hat{\mathbf{x}} \leftarrow \mathcal{D}(\mathbf{z}_T)$ ;           `// frozen decoder`

**15**      $L \leftarrow \mathcal{L}(\hat{\mathbf{x}}, \mathbf{x}_{\text{target}})$ ;           `// loss in pixel space`

**16**      Backpropagate $\nabla_\phi L$ through decoder and all $T$ NCA steps; clip gradients; optimizer step;

     `// Commit evolved latent state to pool`

**17**      Store $(\mathbf{z}_T, \mathbf{c}, \mathbf{x}_{\text{target}})$ in $\mathcal{P}_{\text{pool}}$ (FIFO, capacity-limited);

---

---

**Algorithm 3:** Self-Classification NCA Training

---

**Input** : Training sample $(\mathbf{x}_{\text{input}}, y)$ with class label $y \in \{1, \ldots, K\}$, NCA model $\mathcal{F}_\phi$,
           loss $\mathcal{L}$ (MSE or Cross-Entropy), step range $[T_{\min}, T_{\max}]$, fire rate $p_{\text{fire}}$

**Output:** Trained NCA parameters $\phi^*$

**1** Initialize optimizer, LR scheduler, sample pool $\mathcal{P}_{\text{pool}} \leftarrow \emptyset$;

**2 for** *each training iteration* **do**

**3**      Draw $(\mathbf{x}_{\text{input}}, y)$ from dataset $\mathcal{D}$;

**4**      Construct one-hot target: $\mathbf{y}_{\text{oh}} \in \{0, 1\}^{H \times W \times K}$ broadcast spatially;

**5**      Construct initial state: $\mathbf{s}_0 \leftarrow [\mathbf{x}_{\text{input}}; \mathbf{0}_{C_h}; \mathbf{0}_K]$ ;     `// input (fixed), hidden, class channels`

     `// Sample Pool: optionally replace hidden and class channels with previously`
        `evolved state`

**6**      **if** $|\mathcal{P}_{\text{pool}}| > 0$ **then**

**7**          With probability $r_{pool}$: replace $\mathbf{s}_0$ with a pooled sample;

**8**          Optionally mutate the input image $\mathbf{x}_{\text{input}}$;

     `// Multi-step NCA rollout`

**9**      Sample rollout length $T \sim \mathcal{U}[T_{\min}, T_{\max}]$;

**10**     **for** $t = 0$ **to** $T - 1$ **do**

**11**         $\mathbf{p}_t \leftarrow \mathcal{P}_\phi(\mathbf{s}_t)$ ;                  `// perception: local feature extraction`

**12**         $\Delta\mathbf{s}_t \leftarrow \mathcal{U}_\phi(\mathbf{p}_t)$ ;                   `// update: compute state increment`

**13**         Sample mask $\mathbf{m}_t \sim \text{Bernoulli}(p_{\text{fire}})^{H \times W}$;

**14**         $\mathbf{s}_{t+1} \leftarrow \mathbf{s}_t + \mathbf{m}_t \odot \Delta\mathbf{s}_t$ ;                 `// stochastic update`

**15**         Restore fixed input channels: $\mathbf{s}_{t+1}[: C_{\text{in}}] \leftarrow \mathbf{x}_{\text{input}}$ ;     `// input channels do not evolve`

     `// Loss on classification channels`

**16**     $\mathbf{c}_T \leftarrow \mathbf{s}_T[C_{\text{in}} + C_h :]$ ;              `// extract K classification channels`

**17**     $L \leftarrow \mathcal{L}(\mathbf{c}_T, \mathbf{y}_{\text{oh}})$ ;                 `// pixel-wise MSE or Cross-Entropy`

**18**     Backpropagate $\nabla_\phi L$ through all $T$ steps; clip gradients; optimizer step;

     `// Prediction and pool commit`

**19**     $\hat{y} \leftarrow \arg\max_k \frac{1}{HW} \sum_{i,j} \mathbf{c}_T[i, j, k]$ ;         `// spatial average → argmax`

**20**     Store $(\mathbf{s}_T, y)$ in $\mathcal{P}_{\text{pool}}$ (FIFO, capacity-limited);

---

With SP, the NCAs demonstrated a strong positive recovery (58.74% for damage, 95.75% for mutation), confirming the significant contribution of SP to the robustness of generative NCAs in this setup. These results motivate our use of SP in all subsequent generative experiments.

Table 5: Impact of Sample Pooling (SP) on Regeneration Performance for Emoji Generation, using a 3x3 Convolutional Perception module.

| Training Setup | Damage Regen. (%) | Mutation Regen. (%) |
| --- | --- | --- |
| Without SP | -196.18 | -125.05 |
| With SP | 58.74 | 95.75 |

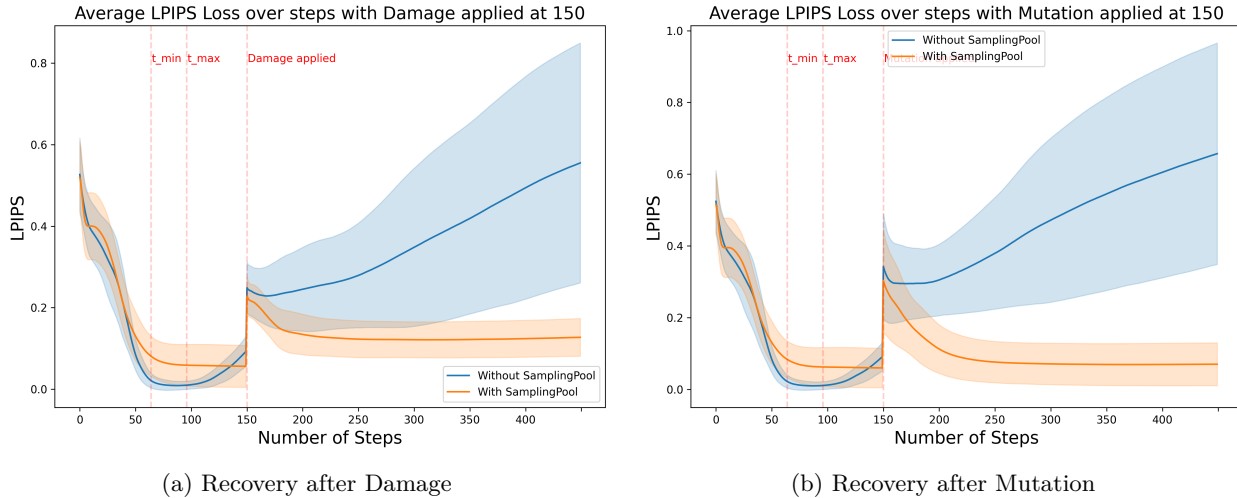

(a) Recovery after Damage

(b) Recovery after Mutation

Figure 18: Comparison of recovery dynamics (e.g., loss curves over time post-perturbation) with and without Sample Pooling (SP) for the Emoji generation task using a 3x3 Conv perception. Demonstrates SP's positive impact on stability.

## A.5 Growing MNIST

**Our Extension.** We also apply this growing paradigm to the MNIST digit generation task, a dataset not explored in the original work, to investigate how different loss functions affect the generative diversity and visual style of the produced patterns. Using a uniform 3x3 convolutional perception module across all experiments, we isolate the effect of the training objective. The NCA autonomously grows complete digits from a single seed pixel, whose color encodes the target digit class (e.g., green for digit 8), with no explicit global condition channels. We systematically compare four loss configurations: MSE, L1 loss, L1 combined with an adversarial critic, and L1 + Critic with Classifier-Free Guidance (CFG) (Ho & Salimans, 2022). Our focus is on understanding whether and how adversarial objectives and guidance mechanisms can introduce controlled stochasticity and visual variation in the generated outputs, beyond the deterministic reconstruction achieved by MSE or L1 alone.

**Results.** Quantitatively, all approaches successfully generate recognizable digits: a pre-trained MNIST classifier achieved 100% accuracy on images produced by NCAs trained with all tested loss functions. However, the qualitative results, visualized in fig. 19, reveal significant differences in generative diversity. Models trained with MSE produce highly deterministic and nearly identical digits across multiple generation runs. In contrast, incorporating an adversarial critic introduces noticeable stylistic variation, and adding CFG leads to substantially greater diversity in digit appearance, as reflected in the pixel variance heatmaps. These findings demonstrate that, while the growing NCA paradigm generalizes well to new datasets like MNIST, the choice of loss function serves as a critical lever to control the trade-off between reconstruction fidelity and generative diversity in NCA-based pattern formation.

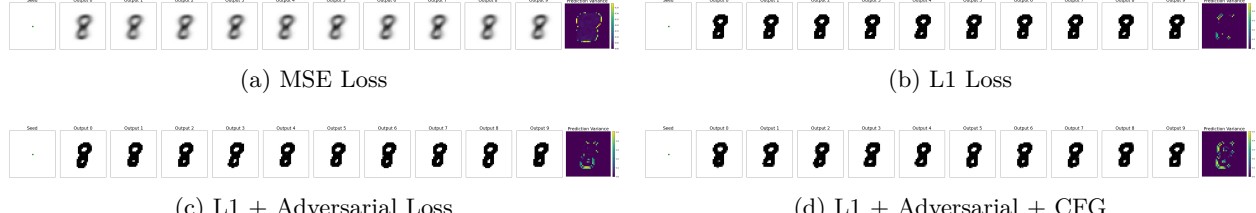

Figure 19: Comparison of generative diversity on the growing MNIST task under different loss functions. Each sub-figure shows the results for a specific loss: (a) MSE, (b) L1, (c) L1 + Adversarial, and (d) L1 + Adversarial + CFG. Each panel displays the initial seed, nine different generated samples, and a heatmap of pixel variance over the samples.

Table 6: Runtime and memory comparison of the evaluated baselines and NCA variants. "Params at inference" counts only parameters active at test time; for the GAN this excludes the critic. "Inference latency" denotes the measured latency of one full forward pass for feed-forward models and one full 32-step rollout for NCA models.

| Model | Params at inference | Train ms/step | Inference latency (ms) | Peak train mem. (MB) Batchsize 16 | Peak infer mem. (MB) Batchsize 64 |
|---|---|---|---|---|---|
| U-Net | 2,768,772 | 3.56 | 1.00 | 195.7 | 375.2 |
| GAN | 2,768,772 | 11.55 | 0.97 | 313.5 | 417.3 |
| NCA (3×3 Conv) | 19,376 | 29.18 | 6.63 | 2813.2 | 352.3 |
| NCA (Attention) | 18,132 | 201.57 | 13.77 | 17444.6 | 2994.4 |

## A.6 Runtime and Memory Benchmark

We report a runtime and memory benchmark for four models used in this paper: a U-Net baseline, a pix2pix GAN, a NCA with $3 \times 3$ perception, and an NCA with attention-based perception. The benchmark was executed on a synthetic image-to-image task at $64 \times 64$ resolution on an NVIDIA RTX 4090. Training used batch size 16 for 200 optimization steps, and inference throughput was measured with batch size 64. For both NCA variants, the model was unrolled for 32 update steps during training and inference. Since the U-Net and GAN are single-pass models, their inference time per step is equal to the latency of one forward pass. For the NCA models, inference time per step is reported as the full rollout with 32 iterations.

Several trends are evident from Table 6. First, parameter count alone is not a reliable measurement for computational cost in the NCA setting. Both NCA variants use orders of magnitude fewer parameters at inference than the U-Net and GAN baselines, yet they incur substantially higher training costs due to the fact that they must be unrolled over many update steps and backpropagated through time. This effect is already visible for the $3 \times 3$ NCA, which uses only 19,376 inference parameters but requires 29.18 ms per training step and 2813.2 MB of peak training memory, compared to 3.56 ms and 195.7 MB for the U-Net. Overall, these results highlight a central trade-off of NCA-based models. NCAs can remain highly parameter-efficient, but this compactness does not automatically translate to runtime efficiency. Instead, recurrent rollout and expensive perception operators shift the dominant bottleneck from parameter count to training memory and iterative computation.

