# OpenReview forum: "A New Kind of Network? Review and Reference Implementation of Neural Cellular Automata"
_TMLR — Accepted by TMLR_

### Review · Reviewer_QcmY · 2025-12-02

**Summary Of Contributions:**

The paper proposes a unified NCA framework, NCAtorch, which modularizes state representation, perception, update rules, and training strategies, and systematically evaluates different perception mechanisms across multiple tasks. It also surveys existing NCA literature, provides a reproducible implementation, and demonstrates how factors such as receptive field size, deformable convolutions, and sample pooling affect model robustness and regeneration.\
Strengths:\
（1）A clear, reproducible reference implementation with practical value for the NCA community.\
（2）Comprehensive multi-task experiments that help elucidate the effects of different perception modules.\
Weaknesses:\
Limited theoretical depth, lack of strong external baselines, and a need for clearer emphasis on the core contributions.

**Audience:**

Yes

**Audience Explanation:**

The paper’s focus on Neural Cellular Automata (NCA), including a systematic review, a unified framework, and multi-task empirical analysis targets a niche but steadily growing area within the TMLR community. Although NCA is not a mainstream deep learning paradigm, it has been gaining attention in several interdisciplinary domains such as generative modeling, differentiable physics, complex system simulation, executable image generation, and self-organizing pattern formation.

**Broader Impact Concerns:**

The Discussion section addresses the technical implications of the work but lacks consideration of broader issues, such as how data distribution biases may affect NCA performance and how the model’s efficient generation and regeneration capabilities could introduce potential ethical risks.

**Claims And Evidence:**

Yes

**Claims Explanation:**

The paper’s main claims include: (1) providing the first systematic review of NCAs, (2) introducing a unified framework and notation, and (3) offering a modular reference implementation with multi-task experiments illustrating the impact of different components. Overall, the experimental results are comprehensive, and the open-source codebase is well developed. However, the work shows some limitations in theoretical justification, though the overall quality remains acceptable.

**Requested Changes:**

(1) Although the authors claim to propose a unified theory, the theoretical content presented in the manuscript is largely a reformulation of existing NCA formulations and a decomposition of modules. The work does not provide rigorous theoretical unification or theoretical proofs, resulting in insufficient theoretical depth.\
(2) In image generation, classification, and other tasks, the evaluation is limited to internal comparisons among different NCA variants. There is no comparison with current state-of-the-art models or classical baselines in these domains, making it difficult to assess the actual advantages of NCAs.\
(3) The manuscript repeatedly emphasizes the computational benefits of latent-space NCAs, yet no quantitative evidence is provided (e.g., training time, parameter counts, memory consumption, or computational complexity). Without concrete measurements, the claimed efficiency advantages are not well supported.\
(4) Section 2.1 introduces five perception mechanisms, including an Attention-based Perception module. However, this module does not appear in the experimental evaluation. The authors are encouraged to include results for this variant, or if excluded for specific reasons, clearly explain the rationale.\
(5) The manuscript mixes a survey of prior work, a framework implementation, engineering refinements, and a wide range of task-level experiments—from emoji generation to texture synthesis, CIFAR-10 classification, and video prediction. While the content is rich, the wide scope makes it difficult to identify a focused central contribution.

---

> ### Author Response · Authors · 2026-03-31
> **Requested Changes by Reviewer QcmY - [Part1]**
>
> **Requested Changes [Part1]:**
>
> **(1)**: ***“Although the authors claim to propose a unified theory, the theoretical content presented in the manuscript is largely a reformulation of existing NCA formulations and a decomposition of modules… .”***
>
> We agree that the initial version of the manuscript has been lacking theoretical depth in some sections. Following the reviewers suggestions,
>
> * We added section 2.2 describing the general training process of NCAs with NCAtorch in detail, including a pseudo code listing of the general training algorithm
>   * For each experiment in the evaluation section, we have added a formal description of the training process and optimization objectives, including formal definition of the used loss loss functions. Pseudo-code listings of the specific training algorithms have been added to the appendix
>   * We have added formal definitions of metrics used in the evaluation
>
> Regarding proofs: all essential properties of NCAs (like the CNN duality) have been proven in literature are cited accordingly. Since the focus of our contribution is the unified modular implementation and systematic empirical evaluation, we don’t see any missing proofs that would constitute an essential gap in our line of argumentation.
>
> **(2)**: ***“In image generation, classification, and other tasks, the evaluation is limited to internal comparisons among different NCA variants. There is no comparison with current state-of-the-art models or classical baselines in these domains… .”***
>
> We sincerely thank the reviewer for this valuable feedback, which has significantly strengthened our manuscript. We have revised the paper to include external baseline comparisons across multiple tasks:
>
> -  Emoji Generation (Table 1): We added U-Net and GAN baselines at three different parameter scales (25k, 44k, and 69k parameters) to enable fair comparison with NCAs of similar complexity.
> - MNIST Classification: We added a GAN baseline with comparable model size achieving 94.78% accuracy, and explicitly referenced state-of-the-art CNNs that routinely achieve near-human performance exceeding 99%. We clarify that MNIST classification is a well-solved problem and our contribution is demonstrating the proof-of-concept, not competing with established methods.
> - CIFAR-10 Classification: We similarly acknowledged that state-of-the-art deep residual networks and Vision Transformers achieve accuracies up to 99.5%, emphasizing that our NCA implementation demonstrates the paradigm's extensibility to complex natural images.
>
> Throughout the revised manuscript (highlighted in red), we have clarified that our primary contribution is the systematic evaluation of architectural components within the NCA paradigm and the demonstration of unique emergent capabilities (regeneration, adaptation through iterative local updates) rather than achieving state-of-the-art benchmark performance.
>
> **(3):** ***“The manuscript repeatedly emphasizes the computational benefits of latent-space NCAs, yet no quantitative evidence is provided...”***
>
> We thank the reviewer for highlighting this gap. We have added a new subsection "Scalability Analysis: Latent vs. Pixel-Space NCAs" (Section 4.1.3) that provides comprehensive quantitative measurements of computational efficiency. We benchmarked latent-space NCAs (using VQVAE with 8× spatial compression) against pixel-space NCAs on an RTX 4090 GPU (24GB VRAM) across multiple resolutions. The analysis reveals that the efficiency gains stem from two factors: (1) the compressed grid size reduces memory quadratically, and (2) proportionally fewer NCA steps are required (e.g., 64 steps for 1024×1024 in latent space vs. 512 steps that would be needed in pixel space). These quantitative results demonstrate that latent-space NCAs are not merely an optimization but a necessary architectural choice for high-resolution generation.

---

> ### Author Response · Authors · 2026-03-31
> **Requested Changes by Reviewer QcmY - [Part2]**
>
> **Requested Changes Part(2):**
>
> **(4)**: ***“Section 2.1 introduces five perception mechanisms, including an Attention-based Perception module. However, this module does not appear in the experimental evaluation…”***
>
> We thank the reviewer for noting this. We have now included the Attention 3x3 perception results in Table 1 (emoji generation task). As the reviewer correctly identified from Section 2.1, this perception module was implemented in our framework but initially excluded from the main results due to training instability.
>
> **(5)**: ***“The manuscript mixes a survey of prior work… . While the content is rich, the wide scope makes it difficult to identify a focused central contribution.”***
>
> We sincerely thank the reviewer for this thoughtful observation. We understand that the breadth of the paper might initially seem scattered, and we appreciate the opportunity to clarify our unified vision. As reflected in our title, "Review and Reference Implementation of Neural Cellular Automata", our work intentionally addresses a critical gap in the NCA research landscape: the lack of standardized, reproducible implementations. Different papers use incompatible codebases, making it difficult to understand which architectural choices generalize across tasks. Our Central Contribution: We provide the first comprehensive, modular reference implementation that enables controlled comparison of NCA components across diverse problem domains.
> Rather than claiming novelty on individual benchmarks, we demonstrate that our framework enables the reproducible, systematic experimentation needed to establish NCA design principles. We believe this positions the paper as a foundational resource, combining review, implementation, and empirical analysis, unified by the goal of accelerating progress in this research area.
>
> **Broader Impact Concerns:**
> ***“The Discussion section addresses the technical implications of the work but lacks consideration of broader issues… .”***
>
> We thank the reviewer for this suggestion. We will add a brief broader-impact paragraph to the Discussion. In particular, we will note that, as with other generative models, the method may inherit biases from the training data distribution and could be misused for synthetic content generation. Since this work is methodological and does not introduce a new application domain, we will keep this discussion brief and focused.

---

### Review · Reviewer_LCbg · 2026-01-11

**Summary Of Contributions:**

The paper presents NCAtorch, a modular PyTorch library for Neural Cellular Automata, accompanied by a review of existing NCA literature and systematic experiments across multiple vision tasks. The authors claim three contributions: (i) the first systematic review of NCA research, (ii) a unified theoretical framework and notation consolidating existing NCA variants, and (iii) an open-source reference implementation.
The experimental evaluation compares various perception modules across image generation, texture synthesis, classification (MNIST, CIFAR-10), and video prediction (MovingMNIST) tasks.

Strengths: Systematic ablation of perception modules; extends NCA evaluation to new tasks; modular software design appears useful and well-structured; visualisation toolkit appears useful.

Weaknesses: Novelty claims are perhaps overstated; no comparison to existing NCA libraries.

**Audience:**

Yes

**Audience Explanation:**

Yes. The software implementation appears well structured, and the systematic evaluation of perception modules across tasks provides useful reference points. Both would serve researchers interested in neural cellular automata.

**Claims And Evidence:**

No

**Claims Explanation:**

The experimental findings (perception ablation, task extensions) and the software library itself are reasonably well supported. However, the three stated contributions are perhaps overclaimed:

(I) "First systematic review": perhaps a bit strong, it seems like Mordvintsev et al. (2020) already consolidates NCA concepts quite similarly to what is presented here. As someone less familiar with NCA beforehand, I liked Section 2 on initial reading, but was surprised that much of the information was already established in prior work.

(II) "Unified theory": The formulation is notational consolidation rather than theoretical novelty. This decomposition seems to be implicit in Mordvintsev et al. (2020). That said, the experimental evaluation does seem useful and clearly presents what has been extended.

(III) "NCAtorch": No comparison to existing libraries is provided to establish what NCAtorch uniquely offers. The software library seems useful and well executed, but it is unclear what the novelty is relative to existing implementations like CAX.


I believe there is value in this paper, primarily the systematic empirical comparison and a modular PyTorch implementation enabling this comparison, which would be useful for other researchers to do further comparisons. However, I am personally a bit less confident in some of the stated novelty claims. Reducing these claims or providing stronger differentiation from prior work would resolve this for me.

Disclaimer: I am open to deferring to other reviewers on this if they think claims are supported. I hold this loosely as I am unfamiliar with NCA specifically and may be missing context.

**Requested Changes:**

Reduce claims:

- Scope "first systematic review" or rephrase to something like "systematic experimental comparison of..." Acknowledge prior consolidation efforts (e.g., Distill series).
- Reframe "unified theory" as notational/pedagogical consolidation. A lot of the modular design breakdown appears fairly standard given Mordvintsev et al. (2020) and others. I would like any distinct new framing in this paper vs Mordvintsev et al. (2020) to be made more clear.
- Specify which "novel details for efficient and stable NCA implementations" are inherited from prior NCA work, which are standard DL engineering, and which are genuinely new.
- Add brief positioning for NCAtorch relative to existing implementations (e.g., CAX, ncalib). A short clarification on what it uniquely offers would suffice.

Minor:
- Use \citet instead of \citep when citations are sentence subjects.
- Grammar: "latent space NCAs. texture generation" needs a comma or semicolon.
- Inconsistent figure references ("Figure" vs "fig.").

---

> ### Author Response · Authors · 2026-03-31
> **Requested Changes by Reviewer LCbg**
>
> **Requested Changes:**
>
> **(1)**: ***“Scope "first systematic review" or rephrase to something like "systematic experimental comparison of..." Acknowledge prior consolidation efforts (e.g., Distill series).”***
>
> We agree that the original phrasing was too strong. We have softened the claim in the contributions section. This acknowledges that prior consolidation efforts exist (notably the Distill series) while clarifying that our contribution is a systematic experimental comparison and implementation of framework across tasks and architectures, which to our knowledge has not been done before.
>
> **(2)**: ***“Reframe "unified theory" as notational/pedagogical consolidation. A lot of the modular design breakdown appears fairly standard given Mordvintsev et al. (2020) and others. I would like any distinct new framing in this paper vs Mordvintsev et al. (2020) to be made more clear.”***
>
> We have replaced all instances of "unified theory" throughout the manuscript. The abstract now reads "unified modular framework and notation" and Contribution II now states: "We introduce a unified notation and modular decomposition that consolidates the disparate formulations of prior NCA works into a single coherent framework." The discussion section has been similarly updated to refer to "technical consolidation and engineering contributions."
>
> **(3)**: ***“Specify which "novel details for efficient and stable NCA implementations" are inherited from prior NCA work, which are standard DL engineering, and which are genuinely new.”***
>
> We have revised the "What NCAtorch adds" paragraph (Section 2\) to explicitly separate these three categories. The text now distinguishes: (a) NCA-established mechanisms inherited from prior work (b) standard deep-learning engineering features and (c) contributions of this work.
>
> **(4):** ***“Add brief positioning for NCAtorch relative to existing implementations (e.g., CAX, ncalib). A short clarification on what it uniquely offers would suffice.”***
>
> We have added a "Related Frameworks" paragraph in Section 2 that explicitly positions NCAtorch relative to existing implementations. A subsequent "What NCAtorch adds" paragraph clarifies what NCAtorch uniquely offers: a PyTorch-centered research framework for controlled, cross-task comparisons with a unified training/evaluation stack, standardized configuration, shared registries, and logging/visualization utilities.
>
> **Minor Requested Changes:**
>
> **(5)**: ***“Use \\citet instead of \\citep when citations are sentence subjects.”***
> Fixed throughout the manuscript. All citations that serve as grammatical subjects now use \\citet.
> **(6)**: ***“Grammar: "latent space NCAs. texture generation" needs a comma or semicolon.”***
> We have revised the sentence structure in the evaluation overview.
> **(7)**: ***“Inconsistent figure references ("Figure" vs "fig.").”***
> We fixed this.

---

### Review · Reviewer_CyDm · 2026-03-18

**Summary Of Contributions:**

The authors build on the idea of a cellular automata to build a CNN based network that learns new transition rules for an automata - designed to execute the given task. The authors have released a PyTorch package to implement it. The description in the text of the process deployed is too thin to draw much of any conclusions.

The authors do seem to do an extensive testing of their implementation of NCA across multiple benchmarks like on standard image classification tasks, difficult video prediction task, texture generation and simple image generation tasks.

**Additional Comments:**

N/A

**Audience:**

Yes

**Audience Explanation:**

This is a new kind of ML and if explained well, its likely to appeal to some parts of the ML audience.

**Claims And Evidence:**

No

**Claims Explanation:**

The list of ``Requested Changes'' makes it explicit as to how much evidence is missing in support of the claims.

**Requested Changes:**

Currently the draft does nothing to teach an external audience/usual ML scientist about NCA. This is a big missing piece that needs to be rectified. NCA is too niche a subject and a standard ML scientist coming from a background of say having read a standard ML book (like https://www.cs.huji.ac.il/~shais/UnderstandingMachineLearning/understanding-machine-learning-theory-algorithms.pdf) has no idea of what is NCA. I suspect that is a majority part of the TMLR audience and that gap hasnt at all been tried to be bridged.

For a start we need a fully algebraic description of the model and its loss function -- something that possibly fits the ``standard" setup as in say equation 3.3 and 3.4 of the above linked book. What is the corresponding expressions here for these 2 basic equations? And if such a thing is not possible then that needs to be clearly explained!

As of now equations 1 and 2 are communicating very little. It neither helps infer the loss or the risk function being trained or the dimensionalities of the variables $s_t$, the domain and range dimensions of the functions ${\cal R}_\phi,{\cal U}_\phi$ and ${\cal P}_{{\cal N},\phi}$ etc. These functions need a rigorous mathematical definition!

Secondly, the method of training of the NCA is also unclear. Its imperative that a pseudocode be given for the training algorithm.

Thirdly, maybe for familiar case of MNIST classification task one can give the exact specification of the NCA loss that purportedly solved that. Its very hard to understand how the dynamical system described in equation 2 becomes a way to solve a classification task. Many pages of necessary mathematics are seemingly missing from  this draft!

Lastly, if we look at things like Table 1 and Figure 7, we see metrics like ``Damage", "Mutation", LIPIPS etc which dont seem to have been defined anywhere.

---

> ### Author Response · Authors · 2026-03-31
> **Requested Changes by Reviewer CyDm**
>
> **Requested Changes:**
>
> **(1)**: ***“For a start we need a fully algebraic description of the model and its loss function \-- something that possibly fits the \`\`standard" setup as in say equation 3.3 and 3.4 of the above linked book. What is the corresponding expressions here for these 2 basic equations? And if such a thing is not possible then that needs to be clearly explained\!”***
>
> Equations (3.3) and (3.4) in the cited book are formal definitions of a classifier. While we agree that readers would certainly benefit from detailed mathematical formulations of NCAs and their training objectives, this needs to be provided for each specific task separately. The NCA architecture is very task generic and we show its usage for generative as well as classification tasks. Following the reviewer's suggestion, we added formal sub-sections to all investigated tasks, providing detailed formulations of the task objectives (e.g. loss functions and evaluation metrics) as well as training details.
>
> **(2)**: ***“As of now equations 1 and 2 are communicating very little. It neither helps infer the loss or the risk function being trained or the dimensionalities of the variables, the domain and range dimensions of the functions ${\cal R}\phi,{\cal U}\phi{\cal P}_{{\cal N},\phi}$ etc. These functions need a rigorous mathematical definition\!”***
>
> Equations 1 and 2 are intended as an abstract description of the overall framework and general NCA structure, not as a complete specification of a single task-specific objective. The precise loss, variable dimensions, and formal operator definitions depend on the particular application. We will make this clearer in the revised manuscript and will provide a formal definition for the actual implementations.
>
> **(3)**: ***“Secondly, the method of training of the NCA is also unclear. Its imperative that a pseudocode be given for the training algorithm.”***
>
> We agree that the training procedure was insufficiently described in the original submission. We have added a new subsection "Training Algorithm" (Section 2.2) with a detailed textual description and full pseudocode (Algorithm 1\) for the standard NCA training loop on generative image tasks. The algorithm shows the key aspects that distinguish NCA training from standard supervised learning. Each experimental section now explicitly references this algorithm and specifies the task-specific loss function used. Additionally, we provide additional algorithms in the appendix (Algorithm 2 and 3\) detailing the latent-space and classification NCA training variant.
>
> **(4)**: ***“Thirdly, maybe for familiar case of MNIST classification task one can give the exact specification of the NCA loss that purportedly solved that. Its very hard to understand how the dynamical system described in equation 2 becomes a way to solve a classification task. Many pages of necessary mathematics are seemingly missing from this draft\!”***
>
> We appreciate this concern and have added a detailed explanation in the revised classification section (Section 3.3). The key insight is that the NCA state is partitioned differently for classification: the input image is placed into fixed (non-evolving) channels, while K additional evolving classification channels (one per class) are initialized to zero. As the NCA iterates via Equation 2, information from the fixed input channels propagates through local perception into these classification channels. The loss is computed between the evolving classification channels and a spatially broadcast one-hot target. This procedure is explicitly described as an adaptation of Algorithm 1, with the two key differences (fixed input channels and loss on classification channels rather than visible channels) clearly stated. The complete classification training pseudocode is provided as Algorithm 3 in the appendix.
>
> **(5)**: ***“Lastly, if we look at things like Table 1 and Figure 7, we see metrics like \`\`Damage", "Mutation", LIPIPS etc which dont seem to have been defined anywhere.”***
>
> Thank you for pointing this out. We have added a dedicated paragraph in Section 3.1 (Regeneration Metric) that formally defines all three metrics before they appear in Table 1 and Figure 7\. Specifically: (1) LPIPS is now defined; (2) Damage is defined as the application of a circular binary mask that zeros out a contiguous region of cells; (3) Mutation is defined as replacing the condition channel with a randomly sampled target class while keeping the cell state intact.

---

### Review · Reviewer_gv4d · 2026-04-06

**Summary Of Contributions:**

This paper claims to provide the "first" systematic review of Nural Cellular Automata (NCA) research, a unified theoretical perspective on NCAs, and a reference implementation of these models and library (NCAtorch) for the community to test these models. More recently the paper has been revised and some of these claims have been adjusted. The main objective of the work is thus to provide a unified notation and PyTorch implementation for NCAs. The experiments focus on several tasks implemented with NCAs, including image generation, classification, video transition, etc. and is thus quite broad. Overall, the authors aim to improve and make NCA research more accessible through this paper and package.

**Strengths:**
1. To my knowledge there exists a genuine gap in terms of implementations of NCAs.
2. It shows that NCAs can be applied to several diverse tasks and the experimental evaluations are quite broad, which is nice to see.
3. Presentation is mostly good but plots and figures need polishing.

**Weaknesses:**

As I am not an expert in NCAs, my review will mostly centre around the implementation itself. I had earlier criticisms about the framing of the claims, i.e., that it was too strong to be called the first review on NCAs given prior work (the overall list of literature cited also seemed quite thin, but I am not aware of much of the NCA literature to comment well here) and the claim of theoretical results, but these have been raised by other reviewers and the others have worked towards addressing them. The supplementary material is no longer visible now but my assessment is based mainly on the initial submission:
1. The authors have not compared their results with NCAs to popular deep learning models for each problem domain, but this would be important to understand current performance gaps and better contextualise NCAs and their assets and limitations.
2. It would be important to include numbers on training and runtime of the latent NCAs to completely assess their efficiency.
3. I think the code could be improved, it does not seem like a proper library with an easy to use or clear API and can be improved a bit:
  * In Section 3.1.4 the authors say they use a VQVAE, but I only found a VAE in the code. Is the implementation provided complete?
  * The code has no tests and is not installable as a package, as there is no `pyproject.toml` or `setup.py`.
  * There seem to be bugs and issues from my reading of the code. For example, `LatentWrapper`'s `encode` function indexes the outputs with 0 but this doesn't seem correct, for `ReconstructionLoss` the function passed in should be a loss module object and not a string (issue in `create_metric`, it is recommended to use separate `GradScaler`s for GANs but the code uses the same `self.scaler`, some numbers and constants such as in the loss implementations are not explained and just hardcoded, etc.
  * Given these issues and the lack of clear and easy to use structure in the package, plus lack of comprehensive documentation (although there are a few guides), I think some work will be required to improve the code so it is easier to use for others as a library.

**Audience:**

Yes

**Audience Explanation:**

The paper provides a reference implementation of NCAs and a survey of existing work on them, in addition to a consistent notation. This could appeal to some of TMLR's audience and is useful for those working on NCAs (particularly the implementation).

**Broader Impact Concerns:**

No concerns that I think are particularly worth addressing here.

**Claims And Evidence:**

No

**Claims Explanation:**

I have listed a few key weaknesses that I believe are important for the authors to address in order to have enough convincing and clear evidence to support the claim of having a clear reference implementation and numbers in comparison to existing widely used methods.

**Requested Changes:**

1. I request the authors to compare the numbers with NCAs to popular deep learning models for each problem, to show what exactly the gap is that can be addressed through research enabled by this framework. (Note: post my review and looking at the authors' responses, it seems this is partially addressed.)
2. It would be good to add runtime comparisons in addition to the existing scalability analyses.
3. A thorough cleanup of the code and improvements to the API design in line with my suggestions would be important.
4. I would request the authors to improve the legibility of the plots and figures, and use PDF/SVG/vector figures instead of the current PNG/rasterised ones.

---

> ### Author Response · Authors · 2026-04-08
> **Requested Changes by Reviewer gv4d  - [Part1]**
>
> **Requested Changes [Part1]:**
>
> **(1): *“I request the authors to compare the numbers with NCAs to popular deep learning models for each problem, to show what exactly the gap is that can be addressed through research enabled by this framework. (Note: post my review and looking at the authors' responses, it seems this is partially addressed.)”***
> We have added (partially based on earlier requests from other reviewers) the following baselines:
>
> * Emoji Generation (Table 1): We added U-Net and GAN baselines at three different parameter scales (25k, 44k, and 69k parameters) to enable fair comparison with NCAs of similar complexity.
> * Emoji Generation (Table 1): We extended the emoji perception comparison to include the missing 3x3 attention-based perception. While it achieves competitive generation quality, it requires careful hyperparameter tuning due to less stable training compared to convolutional approaches.
> * MNIST Classification: We added a GAN baseline with comparable model size achieving 94.78% accuracy, and explicitly referenced state-of-the-art CNNs that routinely achieve near-human performance exceeding 99%. We clarify that MNIST classification is a well-solved problem and our contribution is demonstrating the proof-of-concept, not competing with established methods.
> * CIFAR-10 Classification: We similarly acknowledged that state-of-the-art deep residual networks and Vision Transformers achieve accuracies up to 99.5%, emphasizing that our NCA implementation demonstrates the paradigm's extensibility to complex natural images.
>
> **(2)**: ***“It would be good to add runtime comparisons in addition to the existing scalability analyses.”***
>
> We thank the reviewer for this suggestion. We have added a runtime and memory comparison to the appendix and updated the revised version on OpenReview. The new evaluation confirms that NCAs are parameter-efficient generative models, but incur higher wall-clock time due to their iterative rollout nature. This is an inherent property of NCAs rather than an implementation limitation.

---

> ### Author Response · Authors · 2026-04-08
> **Requested Changes by Reviewer gv4d  - [Part2]**
>
> **Requested Changes [Part 2]:**
>
> **(3):** ***“A thorough cleanup of the code and improvements to the API design in line with my suggestions would be important.”***
>
> We thank the reviewer for the detailed and constructive feedback. We have addressed the points raised and summarise the changes below.
>
> * ***API design and library structure:*** We have refactored all five factory modules (loss, perception, update model, latent encoder, trainer) to dict registries, consistent with the approach used in established frameworks. Each registry is the single source of truth: config validators and tests all derive their valid key sets from it automatically. Adding a new component now requires only implementing the class and adding one entry to the relevant registry.
>
> * ***VQVAE implementation:*** The VQVAE was described in the paper but absent from the submitted code. We apologize for this oversight. We have now added the VQVAE in *nca/core/models/auto\_encoder/vqvae.py* and integrated it fully into the encoder registry, training script, and inference pipeline alongside the existing AE and VAE options.
>
> * ***No tests:*** We have added a comprehensive test suite under tests/ covering all major components. Tests are driven by the factory registries — any new loss, perception, update model, or encoder added to a registry is automatically covered without writing additional test cases. Specifically: *test\_loss\_factory.py*, *test\_perception\_factory.py*, *test\_update\_model\_factory.py*, *test\_latent\_encoder\_factory.py*, *test\_ca\_model.py*, and *test\_ca\_model\_combinations.py* (exhaustive cross-product of all perception × update model combinations, including shape preservation and gradient flow checks).
>
> * ***Package management and reproducibility:*** We migrated from a plain requirements.txt to a uv-based setup with a committed uv.lock file. Anyone cloning the repository and running uv sync obtains an exact reproduction of the experimental environment with a single command.
>
> * ***Not installable as a package:*** We have added a *pyproject.toml* with a proper build system, pinned dependency bounds, and CLI entry points (*ncatorch-train*, *ncatorch-train-ae*, *ncatorch-ui*). The package is installable via *uv sync* for exact reproducibility. The NCAtorch package will be made publicly available upon acceptance of the paper.
>
> * ***Code bugs:*** All reported bugs have been fixed, many thanks for pointing these out\!
>
> * ***Hardcoded constants:*** Previously implicit values are now explicit, named fields in the Pydantic config with defaults matching the original behaviour. Config validators import directly from the registries, so valid values are always in sync with the implementation.
>
> * ***Documentation:*** We have substantially improved the documentation throughout the framework. Docstrings have been added to all key classes, covering configuration, models, and trainers. In addition, all existing extension guides (perception modules, update models, datasets) have been refurbished, and a new guide for implementing custom trainers has been added, covering the two-method extension guide and registry registration.We plan to provide a publicly available hosted documentation site in the future.
>
> We further note that this repository serves as the active foundation for ongoing follow-up work, and we continuously monitor and address issues and refactoring opportunities as they arise.
>
> **(4)**: ***“I would request the authors to improve the legibility of the plots and figures, and use PDF/SVG/vector figures instead of the current PNG/rasterised ones.”***
>
> We thank the reviewer for noting this. We have updated the quality of all figures that could be updated, and the majority are now provided in a vector format (PDF).

---

> ### Comment · Reviewer_gv4d · 2026-04-14
>
> I thank the authors for their response. It seems that the anonymous GitHub link is not working currently, and I see the message "Anonymous GitHub is still processing the repository, it can take several minutes" (however, it does not update). Could the authors check this? I would like to look at the updated code in light of the response before providing further comments and/or revising my judgement.

---

> > ### Author Response · Authors · 2026-04-14
> >
> > Thank you for your time and for pointing this out.
> >
> > We apologize that the anonymized GitHub repository was unavailable. This was not caused by our repository itself, but appears to be an issue with the Anonymous GitHub service. Other users have reported similar issues today as well, as listed in the GitHub issue tracker: https://github.com/tdurieux/anonymous_github/issues?q=is%3Aissue
> >
> > To ensure that the updated code remains accessible, we have also made it available here: https://github.com/nnca88828-pixel/anonymized_NCAtorch
> >
> > We hope this resolves the access issue and allows inspecting the updated code.

---

### Author Response · Authors · 2026-03-31
**General Response**

**General Response**

We would like to thank all reviewers for their constructive feedback. While all reviewers generally appreciate our key contributions of implementing a novel, modular and extensible NCA framework:

* *“A clear, reproducible reference implementation with practical value for the NCA community”* (QcmY)
* *“Modular software design appears useful and well-structured”* (LCbg)

and our systematic and extensive empirical evaluation of current NCA approaches

* *“Comprehensive multi-task experiments that help elucidate the effects of different perception modules.”* (QcmY)

Reviewers uniformly suggested increasing the theoretical depth of the manuscript. We welcome this feedback and have prepared a larger revision, including the following major changes:

* Theoretical depth and formalization:
  * We added section 2.2 describing the general training process of NCAs with NCAtorch in detail, including a pseudo code listing of the general training algorithm
  * For each experiment in the evaluation section, we have added a formal description of the training process and optimization objectives, including formal definition of the used loss functions. Pseudo-code listings of the specific training algorithms have been added to the appendix
  * We have added formal definitions of metrics used in the evaluation
* Contribution and own algorithmic extensions:
  * We extended the related work section and now directly compare NCAtorch to other existing NCA implementations
  * Added new section “What NCAtorch adds” to the introduction and summarization sub-sections to all experimental sections describing our contributions and novel extensions in detail
* Scalability, runtime and compute demands
  * We added the new section 3.1.5, providing in-depth run-time and memory demand evaluations benchmarking latent-space vs. pixel-space NCAs.
  * A dedicated runtime and memory comparison has been added to the appendix, confirming that NCAs are parameter-efficient, but incur higher wall-clock time due to their iterative rollout nature.
* New experiments and baseline comparisons:
  * Emoji Generation (Table 1): We added U-Net and GAN baselines at three different parameter scales (25k, 44k, and 69k parameters) to enable fair comparison with NCAs of similar complexity.
  * Emoji Generation (Table 1): We extended the emoji perception comparison to include the missing 3x3 attention-based perception. While it achieves competitive generation quality, it requires careful hyperparameter tuning due to less stable training compared to convolutional approaches.
  * MNIST Classification: We added a GAN baseline with comparable model size achieving 94.78% accuracy, and explicitly referenced state-of-the-art CNNs that routinely achieve near-human performance exceeding 99%. We clarify that MNIST classification is a well-solved problem and our contribution is demonstrating the proof-of-concept, not competing with established methods.
  * CIFAR-10 Classification: We similarly acknowledged that state-of-the-art deep residual networks and Vision Transformers achieve accuracies up to 99.5%, emphasizing that our NCA implementation demonstrates the paradigm's extensibility to complex natural images.
* API design, code quality, and reproducibility:
  * We have refactored all factory modules to dict-based registries.
  * Added a comprehensive test suite driven by these registries.
  * Migrated to a uv-based package setup with a committed lock file.
  * Added a proper pyproject.toml with CLI entry points.
  * Fixed all reported bugs
  * Improved documentation including docstrings for all key classes and new extension guides.
  * The NCAtorch package will be made publicly available upon acceptance.
* All figures that could be updated have been revised, and the majority are now provided in vector format (PDF).


* We added a “broader impact statement” to the discussion section

All of these changes (and many smaller ones not listed here) are marked in red in the manuscript.

Given these changes, we think that we have been able to address the reviewer feedback. We would like to emphasize that the key contribution of this paper is a novel unified, modular and extensible implementation of NCAs and an extensive systematic empirical evaluation of the current state of NCA methods. Our aim is to foster NCA research and to provide the necessary tool and starting point for the community.
We acknowledge that there are many open and exciting theoretical properties of NCAs which should be explored. However, this is not the scope of this (already quite extensive) paper.

---

### Author Response · Authors · 2026-04-08
**Comment to all reviewers:**

We would like to inform the reviewers that, after receiving an additional fourth review, we have updated both our general response and the revised manuscript. We have also added a specific response addressing the new reviewer’s comments.

For transparency, some changes in the manuscript and in the general response were made after our initial submission of responses. We therefore kindly ask all reviewers to consult the most recent version of the paper and the updated general response.

We thank all reviewers again for their time and constructive feedback!

---

### Decision · Action_Editor_nNdo · 2026-05-08

**Recommendation:** Accept with minor revision

**Additional Comments:**

The authors are strongly encouraged to implement all the suggestions of the reviewers in the software package to ensure it is indeed of a high enough standard to enable widespread adoption.

**Audience:**

Yes

**Audience Explanation:**

All reviewers have agreed that the presentation of the framework in the paper would be interesting for the community.

**Claims And Evidence:**

Yes

**Claims Explanation:**

All reviewers agree now that the findings are supported by the evidence, in particular, the reviewers that shared some doubts about the validity of the software package are now satisfied with its quality.